# Redox controls RecA protein activity via reversible oxidation of its methionine residues

**Camille Henry[1,2], Laurent Loiseau[1], Alexandra Vergnes[1], Didier Vertommen[3], Angela Mérida-Floriano[4], Sindhu Chitteni-Pattu[2], Elizabeth A Wood[2], Josep Casadesús[4], Michael M Cox[2], Frédéric Barras[1,5,6]\*, Benjamin Ezraty[1]\***

[1]Aix-Marseille Univ, CNRS, Laboratoire de Chimie Bactérienne, Institut de Microbiologie de la Méditerranée, Marseille, France; [2]Department of Biochemistry, University of Wisconsin-Madison, Wisconsin-Madison, United States; [3]de Duve Institute, Université Catholique de Louvain, Brussels, Belgium; [4]Departamento de Genética, Universidad de Sevilla, Sevilla, Spain; [5]Institut Pasteur, Département de Microbiologie, SAMe Unit, Paris, France; [6]UMR CNRS-Institut Pasteur 2001 Integrated and Molecular Microbiology (IMM), Paris, France

**Abstract** Reactive oxygen species (ROS) cause damage to DNA and proteins. Here, we report that the RecA recombinase is itself oxidized by ROS. Genetic and biochemical analyses revealed that oxidation of RecA altered its DNA repair and DNA recombination activities. Mass spectrometry analysis showed that exposure to ROS converted four out of nine Met residues of RecA to methionine sulfoxide. Mimicking oxidation of Met35 by changing it for Gln caused complete loss of function, whereas mimicking oxidation of Met164 resulted in constitutive SOS activation and loss of recombination activity. Yet, all ROS-induced alterations of RecA activity were suppressed by methionine sulfoxide reductases MsrA and MsrB. These findings indicate that under oxidative stress MsrA/B is needed for RecA homeostasis control. The implication is that, besides damaging DNA structure directly, ROS prevent repair of DNA damage by hampering RecA activity.

**\*For correspondence:**
frederic.barras@pasteur.fr (FB);
ezraty@imm.cnrs.fr (BE)

**Competing interests:** The authors declare that no competing interests exist.

## Introduction

Aerobic metabolism produces reactive oxygen species (ROS) such as superoxide ($O_2^{\cdot-}$), hydroxyl radicals (HO$^\circ$), and the non-radical molecule hydrogen peroxide ($H_2O_2$). Besides the endogenous production, ROS arises from various external sources such as metal, UV radiation, or pathogen defenses (*Imlay, 2019*). ROS are extremely harmful for the cell, leading to nucleic acid damages, protein oxidation, and lipid peroxidation. Exposure to ROS creates enough damage to impair cellular homeostasis in all living systems (*Imlay, 2015*). Within proteins, sulfur-containing cysteinyl and methionyl residues can be oxidized to sulfenic acid adducts and methionine sulfoxide, respectively (*Ezraty et al., 2017*). These post-translational oxidations can be reversed by dedicated protein repair pathways present in all domains of life. Thioredoxin/glutaredoxin reductases can reduce oxidized cysteinyl while methionine sulfoxide reductase (Msr) reduces methionine sulfoxide residues (Met-O).

In prokaryotes, MsrA and MsrB are mostly present in the cytoplasm (*Ezraty et al., 2017*; *Dos et al., 2018*). MsrA and MsrB have a strict stereospecificity for their substrates, with MsrA reducing only Met-*S*-SO and MsrB reducing only Met-*R*-SO diastereoisomers. Consequently, full repair of a given cytosolic oxidized protein requires the action of both MsrA and MsrB (*Tsvetkov et al., 2005*). Lack of functional MsrA/B yielded to pleiotropic phenotypes related to bacterial virulence or aging. Global proteomic or dedicated studies identified substrates, like the

ubiquitous protein SRP54 (*Ezraty et al., 2004*; *Gennaris et al., 2015*) and several other proteins listed in the database MetOSite (https://metosite.uma.es) (*Valverde et al., 2019*). However, a molecular understanding of the contribution of Msr as an anti-ROS defense needs the actual identification of proteins, whose activities are hampered by oxidation and can be rescued by Msr's repairing activity.

In this study, we identified the ubiquitous recombinase RecA as a substrate of MsrA/B. RecA promotes DNA recombination and coordinates stress response by inducing the expression of the SOS regulatory system. We report that RecA is targeted by ROS and this abrogates both RecA recombination and SOS induction capacities. We show that MsrA/B are able to rescue oxidized RecA activities. Furthermore, an in-depth analysis allowed us to show that Met residues are not all oxidized at the same rate, and that functional consequences of oxidation depend upon which Met residue is being hit. Overall we discuss a model that predicts how under oxidative stress both protein repair and *de novo* synthesis are used by *Escherichia coli* to replenish the pool of functional RecA above a threshold value.

## Results

### MsrA/B are required for RecA-dependent viability under stress conditions

An *E. coli* strain derivative lacking all peroxide-scavenging activities (catalases KatE and KatG, and peroxidase AhpC), referred to as Hpx⁻, accumulates micromolar amounts of $H_2O_2$ (*Seaver and Imlay, 2001*). We found the Hpx⁻ *msrA msrB* strain to have significantly reduced viability upon UV stress compared to the Hpx⁻ parental strain with around 1.4 log difference at 50 and 60 $J/m^2$ (*Figure 1A*). UV defect is a hallmark of DNA recombination and/or repair defect. *E. coli* fights UV-induced DNA damages by both a RecA-dependent homologous recombination (HR) pathway and a RecA-independent UvrABCD-dependent nucleotide excision repair (NER) pathway. Therefore, we deleted either the *recA* or the *uvrA* gene in the Hpx⁻ background. In fact, the *recA* mutation proved to be lethal in the Hpx⁻ background in aerobic conditions as previously reported (*Park et al., 2005*). In contrast, the Hpx⁻ *uvrA* strain exhibited no phenotype, presumably because the HR pathway was sufficient to afford UV resistance. We then used Hpx⁻ *uvrA* strain to delete both *msrA* and *msrB* genes. The resulting Hpx⁻ *msrA msrB uvrA* strain showed hypersensitivity to $O_2$, similar to the Hpx⁻ *recA* strain (*Figure 1B*). Altogether, these observations suggested that in the presence of a higher-than-normal level of $H_2O_2$ RecA protein might be damaged by oxidation. MsrA/B thus could be essential for maintaining a level of functional reduced RecA above a threshold value. Consistently, we observed that high *recA* gene dosage suppressed UV sensitivity of the Hpx *msrA msrB* mutant (*Figure 1C*) and the $O_2$ sensitivity of the Hpx⁻ *msrA msrB uvrA* (*Figure 1D*).

### MsrA/B are important for RecA function under oxidative stress *in vivo*

RecA is a multifunctional protein serving in HR and SOS response regulation (*Clark and Margulies, 1965*; *Slilaty and Little, 1987*). We used P1 phage-mediated transduction (*Miller, 1972*) to assess RecA-dependent HR efficiency under oxidative stress. When using the Hpx⁻ *msrA msrB* strain as a recipient, the transductant frequency dropped tenfold compared with the parental MsrA/B-proficient strain (*Figure 2A*). Transductant frequency was restored to the parental strain level by bringing in either plasmid-borne *msrA* or *msrB* genes (*Figure 2A*). Next, we investigated the influence of MsrA/B on the RecA-dependent SOS response. We used a *recA-gfp* gene fusion to monitor the SOS response. The Hpx⁻ strain exhibited enhanced SOS induction compared with the wild-type (WT) strain (*Figure 2B*). This was consistent with a previous work reporting increased DNA damage level in the Hpx⁻ background (*Courcelle et al., 2001*; *Mancini and Imlay, 2015*). When *msrA msrB* deletions were introduced into the Hpx⁻ background, the level of SOS induction decreased and was similar to that observed in the WT strain (*Figure 2B*). The well-known SOS-inducing compound mitomycin C (MMC) was then added. The Hpx⁻ strain exhibited an enhanced level of *recA-gfp* expression compared with the WT (*Figure 2B*). In contrast, deleting *msrA msrB* genes in the Hpx⁻ strain impaired MMC induction of *recA-gfp* (*Figure 2B*). Altogether, these observations showed that under oxidative stress MsrA/B maintains a level of reduced functional RecA necessary to carry out both efficient recombination and SOS regulation.

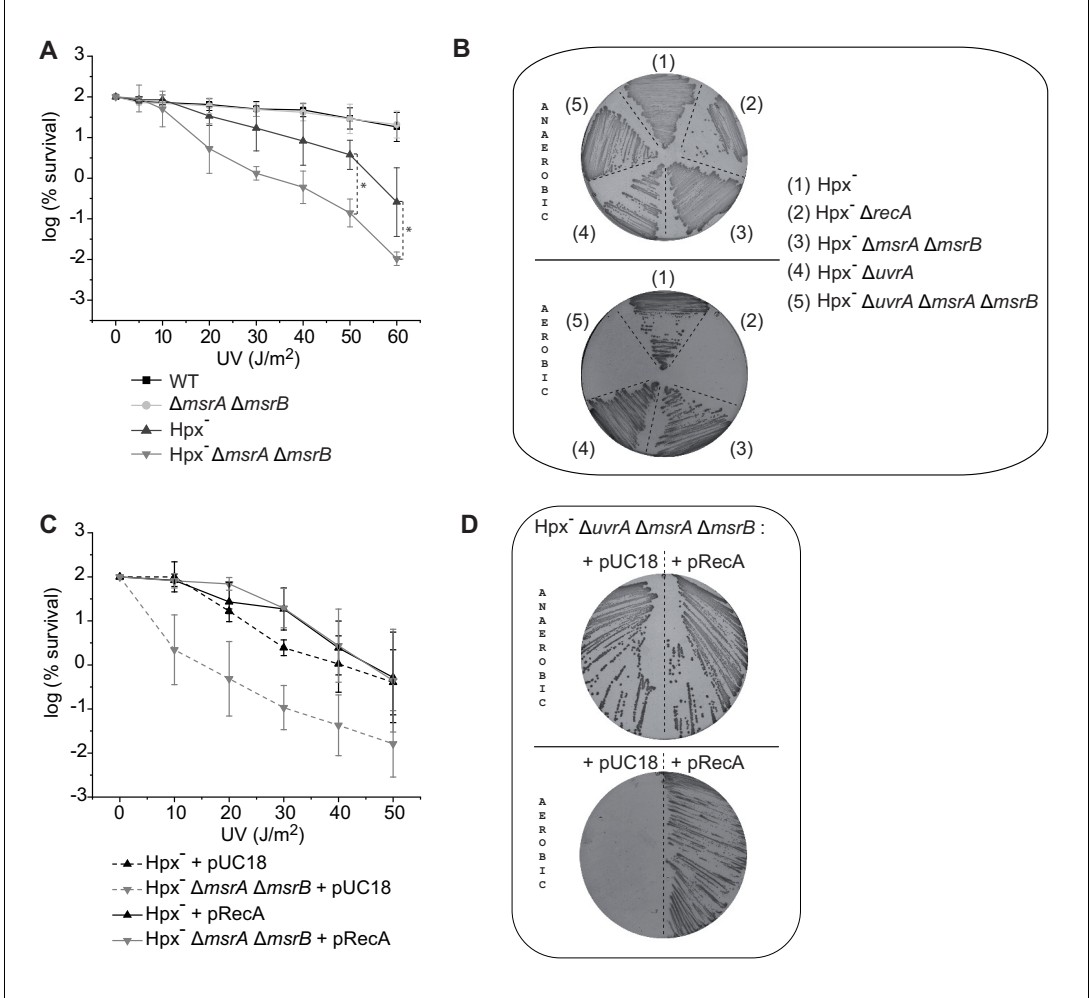

**Figure 1.** MsrA/B essential for maintaining RecA functional. (A) UV sensitivity of wild-type (WT) and mutant strains (BE152 [ΔmsrA ΔmsrB]; BE007 [Hpx⁻]; BE080 [Hpx⁻ΔmsrA ΔmsrB]) was tested by monitoring colony-forming units at different UV doses. Curves represent the mean value of biological triplicates, and error bar ± represents the s.d. (B) Mutated strains (BE007 [Hpx⁻]; LL1609 [Hpx⁻ΔrecA]; BE080 [Hpx⁻ΔmsrA ΔmsrB]; BE032 [Hpx⁻ΔuvrA]; BE033 [Hpx⁻ΔmsrA ΔmsrB ΔuvrA]) on Lysogeny Broth (LB) plates incubated in the presence or absence of oxygen as indicated on the pictures. (C) Suppression assay for UV sensitivity with Hpx⁻ΔmsrA ΔmsrB carrying pRecA. (D) Suppression assay for oxygen sensitivity of BE033 (Hpx⁻ΔmsrA ΔmsrB ΔuvrA) mutant carrying the plasmid pRecA that overexpress RecA. Asterisks indicate a statistically significant difference between Hpx⁻ and Hpx⁻ΔmsrA ΔmsrB, *p≤0.05 (Mann–Whitney U test).

## RecA is subject to reversible methionine oxidation *in vitro* and *in vivo*

We then characterized the effect of oxidative agents on the RecA protein. First, purified RecA protein was treated with hydrogen peroxide ($H_2O_2$) (50 mM) for 2 hr and analyzed by mass spectrometry. On the nine Met residues present in RecA, four could be identified within peptides detectable by mass spectrometry after peptidase digestion. In contrast, peptides containing Met28 and Met197 were too small to be detected. Likewise, Met35, Met58, Met164, and Met202 were found to have around 20% of oxidation (*Figure 3A*). This basal level of oxidation could be explained by the use of in-gel digestion, a technique known to cause Met oxidation (*Klont et al., 2018*), prior to mass spectrometry analysis. After $H_2O_2$ treatment, the level of oxidation increased to approximately 90% at Met164- and Met202-containing positions (*Figure 3A*). Met35 and Met58, being part of the same peptide, we were unable to differentiate the oxidation level of each residue separately. Importantly, no Cys residue was found to be modified. Second, the RecA oxidized above, referred to as RecAox, was treated with MsrA/B enzymes in the presence of 1,4-dithiothreitol (DTT) to yield to RecArep. Mass spectrometry analysis of RecArep revealed a drastic decrease in its content of Met-O residues (*Figure 3A*). As explained above, gel electrophoresis per se might have induced oxidation of Met

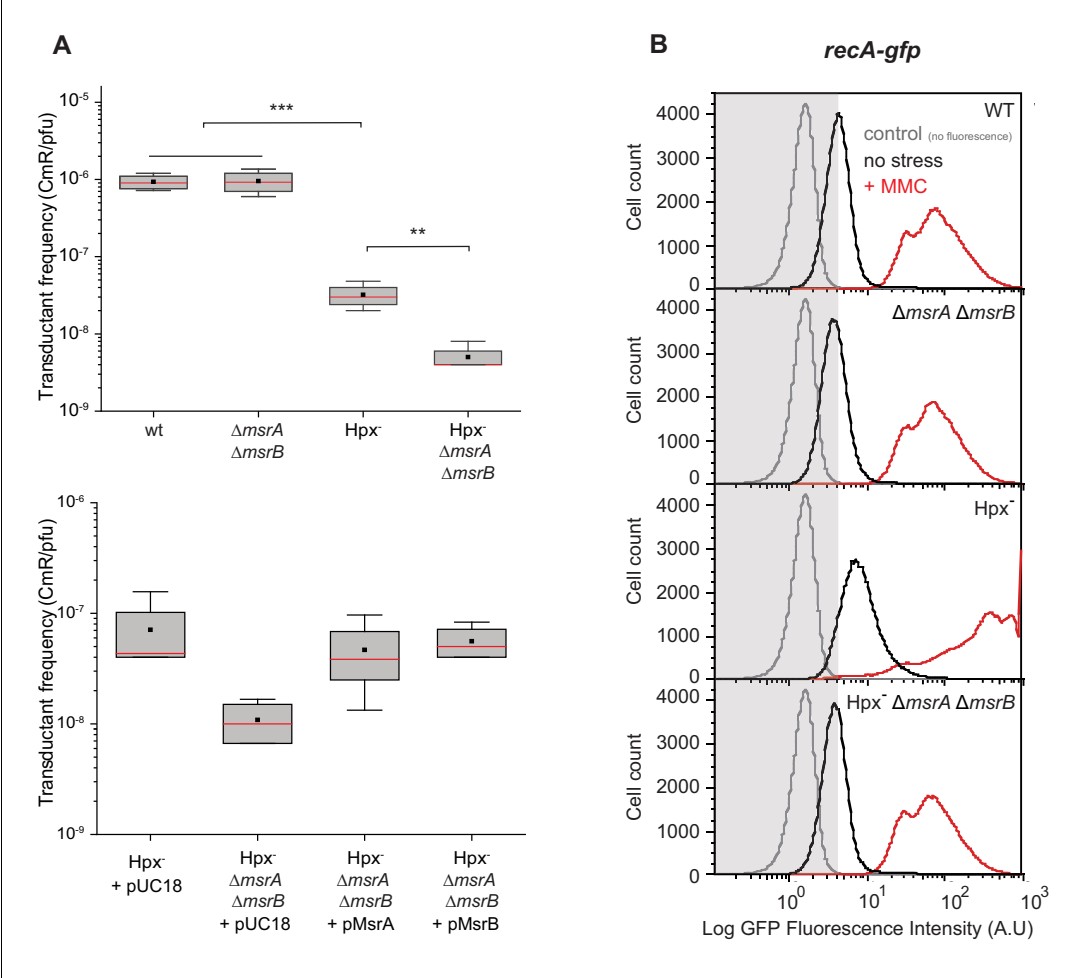

**Figure 2.** Physiological importance of RecA repair by MsrA and MsrB. (**A**) Transductant frequency of a *lacZ-cat* marker in different genetics backgrounds. Top panel shows the transductant frequency of the BE080 (Hpx⁻ΔmsrA ΔmsrB) strain as recipient compared with the BE007 (Hpx⁻) strain, n = 6 and p=0.01. Bottom panel shows the Hpx⁻ and Hpx⁻ΔmsrA ΔmsrB recipient strains carrying an empty vector pUC18, pMsrA, or pMsrB, n = 3–6, p=0.05. (**B**) SOS gene expression at the single cell level, monitored by flow cytometry. Green Fluorescent Protein (GFP) intensity of a transcriptional *recA-gfp* fusion in wild type (WT), ΔmsrA ΔmsrB, Hpx⁻, and Hpx⁻ ΔmsrA ΔmsrB backgrounds cultivated in the absence (black line) or presence of mitomycin C (MMC) (0.25 μg/mL) (red line). The fluorescence intensity is proportional to the *recA* gene expression, which is used as an indicator of SOS response induction. In total, 100,000 events were analyzed for each sample.

residues, and therefore, one must be cautious when considering the quantitation of Met-O as their source might be multiple. However, besides allowing us to identify those Met residues that are sensitive to oxidation and eventually can be reduced by the MsrA/B enzymes, these *in vitro* characterizations allowed us to guide our *in vivo* studies presented later in this article. Third, we used a gel shift assay that detects Met-O residues in a polypeptide as Met-O-containing polypeptides run slower than their reduced counterpart on SDS-polyacrylamide gel electrophoresis (SDS-PAGE) (*Gennaris et al., 2015*). By western blot, we observed that RecA produced in a *msrA msrB* mutant exposed to HOCl treatment run slower than RecA produced in a WT strain or in a *msrA msrB* mutant that has not been exposed to HOCl (*Figure 3B*). These results showed that RecA is subject to oxidation both *in vitro* and *in vivo*, and that in both contexts MsrA/B allows reduction of Met-O residues present in RecAox.

### *In vitro* RecAox is inactive but can be rescued by MsrA/B

By using electron microscopy (*Lusetti et al., 2003b*), we observed that, in the presence of single-strand binding protein (SSB), RecA formed extended filaments on circular single strand (css) DNA (*Figure 3—figure supplement 1*). RecAox was unable to do so (*Figure 3C*), whereas RecArep

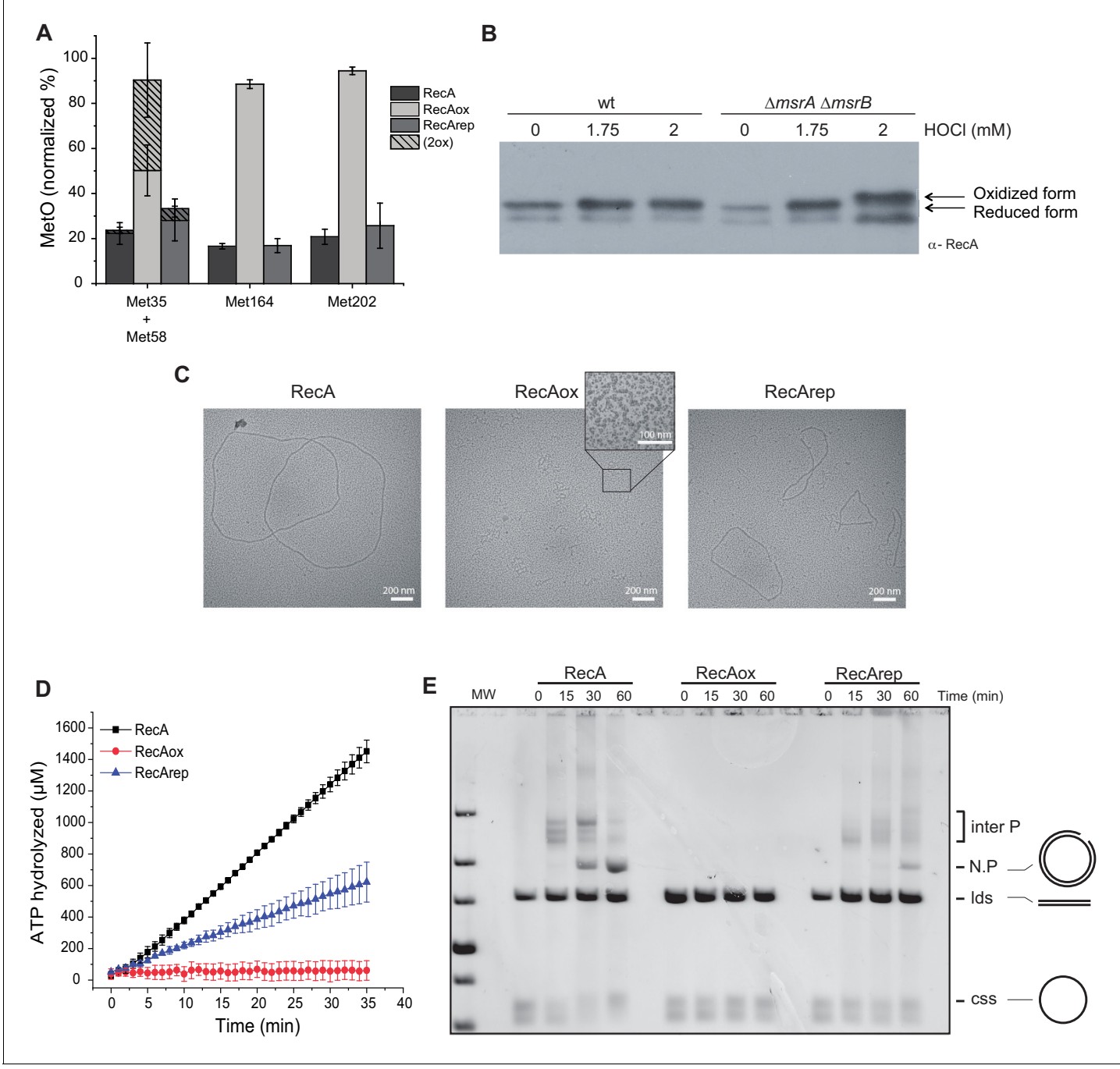

**Figure 3.** Reversible inactivation of RecA by MsrA and MsrB. (**A**) RecA was treated with $H_2O_2$ to induce Met-O formation. Subsequently, oxidized RecA was incubated with MsrA and MsrB proteins in the presence of the reducing agent 1,4-dithiothreitol (DTT). The relative percentage of Met-O in various forms of RecA (native, oxidized [RecAox], and repaired [RecArep]) was determined by mass spectrometry analysis. Error ± means s.d., n = 3. (**B**) Wild type (WT) and BE152 (ΔmsrA ΔmsrB) were cultivated in LB, and HOCl (1.75 and 2 mM) was added. Immunoblot analysis reveals RecA mobility on SDS-polyacrylamide gel electrophoresis. (**C**) Electron microscopy of RecA filaments. Filament formation was studied using different forms of RecA (native, RecAox, and RecArep) as indicated on the top of the pictures. (**D**) ATP hydrolysis rates by RecA, RecAox, and RecArep. (**E**) DNA strand exchange promoted by the different RecA species. The intermediate joint molecules (Inter P) product of exchange between circular single strand (css) and linear double strand (lds) DNA contains a three-stranded branch point that migrates along the molecule until nicked, forming a nicked circular duplex DNA (N.P.).

The online version of this article includes the following figure supplement(s) for figure 3:

**Figure supplement 1.** Quantification analysis of data from *Figure 3C, E*.

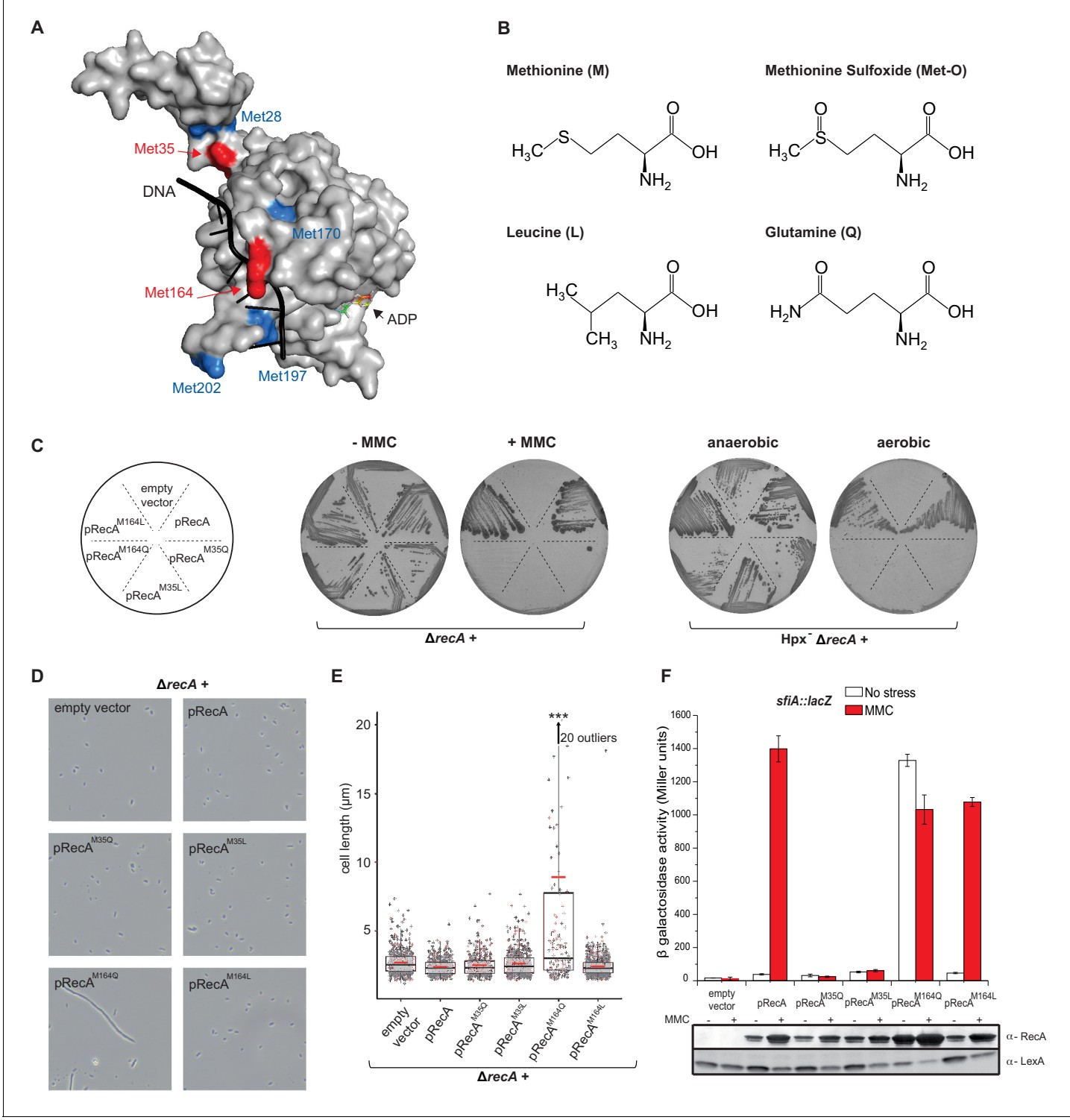

**Figure 4.** Mimicking Met oxidation yields to diverse functional consequences. (**A**) Structure of the *E. coli* RecA-DNA complex adapted from PDB: 3CMW (*Chen et al., 2008*). The cartoon shows the structure of a RecA monomer (surface representation) in the presence of ssDNA in black and the ATP analogue adenosine diphosphate (ADP)-AlF4 binding site (for simplicity, only the structure of ADP is represented). Exposed methionine (Met) residues are colored in blue (Met28, Met170, Met197, Met202) or red (Met35, Met164). (**B**) Diagram of the structure of methionine (M), methionine sulfoxide (Met-O), leucine (L), and glutamine (Q). (**C**) Left panel shows the growth of LL1594 (*ΔrecA*) carrying plasmids with the different *recA* alleles under mitomycin C (MMC) stress. Right panel shows the growth of LL1609 (Hpx⁻*ΔrecA*) carrying the different *recA* alleles under aerobic condition. (**D**) LL1594 (*ΔrecA*) cells expressing *in trans* the different RecA variants were imaged in stationary phase by phase-contrast microscopy. While *ΔrecA*

*Figure 4 continued on next page*

*Figure 4 continued*

expressing RecA, RecA$^{M35L}$, RecA$^{M35Q}$, or RecA$^{M164L}$ exhibit a normal cell shape, long filaments were observed in the presence of RecA$^{M164Q}$. (E) Quantitation of the cell length from the experiment shown in (D). This analysis was carried out on a large number of cells from at least 10 fields. The range of the number of cells is 151–554. Parametric statistical tests (t test) yield high significant statistic (p<10$^{-5}$) between the Δ*recA* + pRecA$^{M164Q}$ group and any of the other groups. Among all the other groups, the differences are not significant. (F) SOS induction after 2 hr exposure to MMC in a Δ*recA* strain expressing *in trans* the different *recA* alleles was monitored by *sfiA::lacZ* expression (top panel) and western blot anti-RecA and anti-LexA analyses (bottom panel). In the wild-type (WT) situation, MMC exposure leads to LexA cleavage, RecA production, and *sfiA* expression. Histogram graph of the mean expression-level value of the biological triplicates (top panel), error bar ± represents the s.d.

recovered part of its ability to form nucleoprotein filaments in the presence of SSB (*Figure 3—figure supplement 1*). Using enzymatic assays, we showed that, as previously reported (*Kowalczykowski and Krupp, 1987*), RecA exhibited DNA-dependent ATPase activity in the presence of SSB with an apparent kcat of 25.22 μM·min$^{-1}$. In contrast, RecAox was unable to hydrolyze ATP (SSB) with an apparent kcat of 0.26 μM·min$^{-1}$, whereas RecArep recovered a significant level of activity with an apparent kcat of 9.66 μM·min$^{-1}$ (*Figure 3D*). RecA is also known to promote DNA strand exchange (*Cox and Lehman, 1982*; *Lindsley and Cox, 1990*). We compared the capacities of RecA, RecAox, and RecArep to promote strand exchange between cssDNA and linear double strand (lds) DNA in the presence of SSB. Over time, formation of nicked circular product (N.P.) was observed in the RecA-containing sample but not in the RecAox-containing reaction (*Figure 3—figure supplement 1*). RecArep was also able to promote DNA strand exchange, though with reduced efficiency compared to RecA (*Figure 3—figure supplement 1*). Together, our results show that RecAox protein cannot catalyze HR, which MsrA/B is partially able to restore.

## RecA$^{M35Q}$, a proxy for oxidation of Met35, hampers RecA activity

Met35, which is targeted by oxidation, is located at the monomer–monomer interface in the RecA proto-filament (*Figure 4A*; *McGrew and Knight, 2003*; *Chen et al., 2008*). We replaced Met35 by either Gln (Q), a mimetic of Met-O, to test the consequences of oxidizing position 35, or Leu (L), to test the physicochemical constraints prevailing at that position (*Figure 4B*; *Drazic et al., 2013*). We reasoned that the variant containing Leu would inform us on the physicochemical constraints prevailing at position 35, and the variant containing Gln would inform us specifically on the effect of oxidation of Met35. To test activities of RecA variants, we used sensitivities to MMC and O$_2$ as well as induction of the SOS *sfiA* gene as readouts. Expression of RecA$^{M35L}$ and RecA$^{M35Q}$ variants was unable to rescue hypersensitivities of Δ*recA* and Hpx$^-$ Δ*recA* strains to MMC and O$_2$, respectively (*Figure 4C*). Moreover, the SOS response was abrogated in the Δ*recA* strain harboring RecA$^{M35L}$ and RecA$^{M35Q}$ (*Figure 4D–F*). We performed biochemical analyses of the RecA$^{M35L}$ variant and found that it had lost ATPase activity, nucleofilament formation, and DNA strand exchange capacities (*Figure 5*; *Figure 5—figure supplement 1*). Altogether, these studies indicated that position 35 is functionally important, with little tolerance for substitution at this position. Surprisingly, such an apparently essential role of Met35 was overlooked in the previous multiple mutagenesis studies (*McGrew and Knight, 2003*). Overall, these results did not allow us to evidence specific effect of oxidation of Met35 residue, besides the fact that it would be, as any other modification of that position, highly detrimental for both DNA repair and SOS regulation.

## RecA$^{M164Q}$, a proxy for oxidation of Met164, stimulates the SOS response

Met164, identified above as a target of oxidation (*Figure 3A*), is located in the L1 region that interacts with DNA (*Figure 4A*; *McGrew and Knight, 2003*; *Chen et al., 2008*). Applying the same logic as for the study of position 35, we carried out phenotypic analyses to investigate the functionality of Gln- or Leu-containing variants at position 164. Expression of the RecA$^{M164Q}$ variant failed to complement hypersensitivity to MMC of Δ*recA* strain (*Figure 4C*). In contrast, expression of the RecA$^{M164L}$ variant allowed full complementation (*Figure 4C*). Similarly, the Hpx$^-$ Δ*recA* strain was hypersensitive to O$_2$ when expressing RecA$^{M164Q}$ but fully viable when expressing RecA$^{M164L}$ (*Figure 4C*). The SOS response was followed by monitoring *sfiA::lacZ* expression, RecA steady-state level and LexA cleavage by immunoblotting, and cellular filamentation by phase contrast microscopy. A strain synthesizing RecA$^{M164L}$ exhibited WT-like features of MMC-induced SOS response

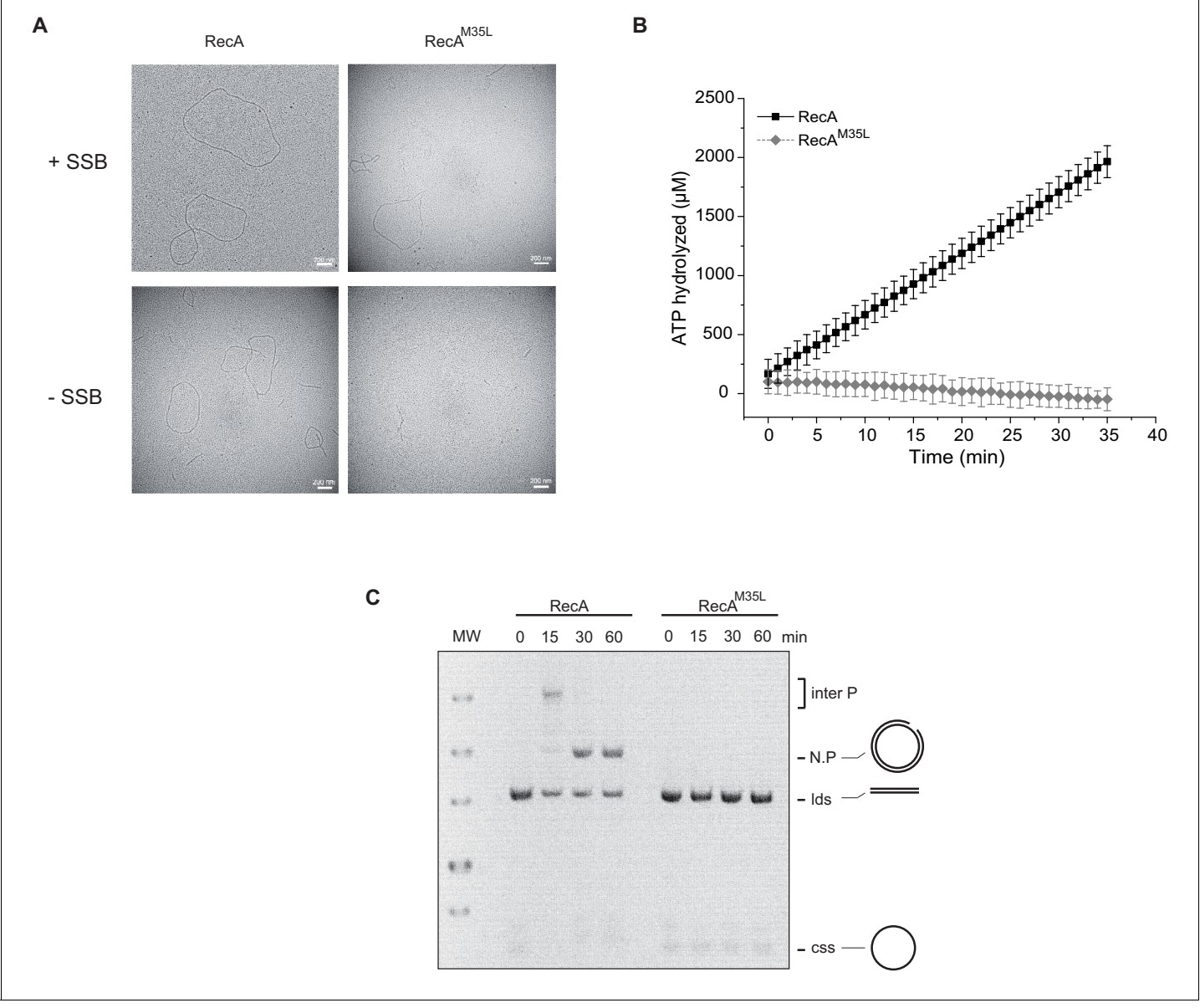

**Figure 5.** Importance of the residue Met35 of RecA. (**A**) Electron microscopy of RecA filament. Filament formation was studied using RecA and RecA[M35L] as indicated at the top of the pictures. (**B**) RecA[M35L] was unable to hydrolyze ATP. (**C**) RecA[M35L] was unable to invade and/or catalyze DNA strand exchange.

The online version of this article includes the following figure supplement(s) for figure 5:

**Figure supplement 1.** Quantification analysis of data from *Figure 5*.

within all tests listed above (*Figure 4D–F*). Surprisingly, a strain synthesizing RecA[M164Q] showed constitutive high levels of *sfiA::lacZ* expression, a high steady-state level of RecA, decreased levels of LexA, and strong cellular filamentation even in the absence of MMC (*Figure 4D–F*). This phenotype was surprising and somehow echoed the long-studied *recA730* allele (RecA[E38K]) as both alleles turn on the SOS response constitutively (*Witkin et al., 1982*). In contrast, *recA730* bears no phenotype regarding MMC sensitivity (*Figure 6A*). In order to better compare both alleles, we introduced the *recA*-M164Q allele in the chromosome. The resulting strain recapitulated all the phenotypes observed with a plasmid encoded RecA[M164Q] version (*Figure 6A, B*). First, it showed the expression of the *sfiA::lacZ* fusion even in the absence of MMC, that is, it was constitutive for SOS activation. Second, it proved to be sensitive to MMC treatment, indicating that RecA[M164Q] protein is deficient

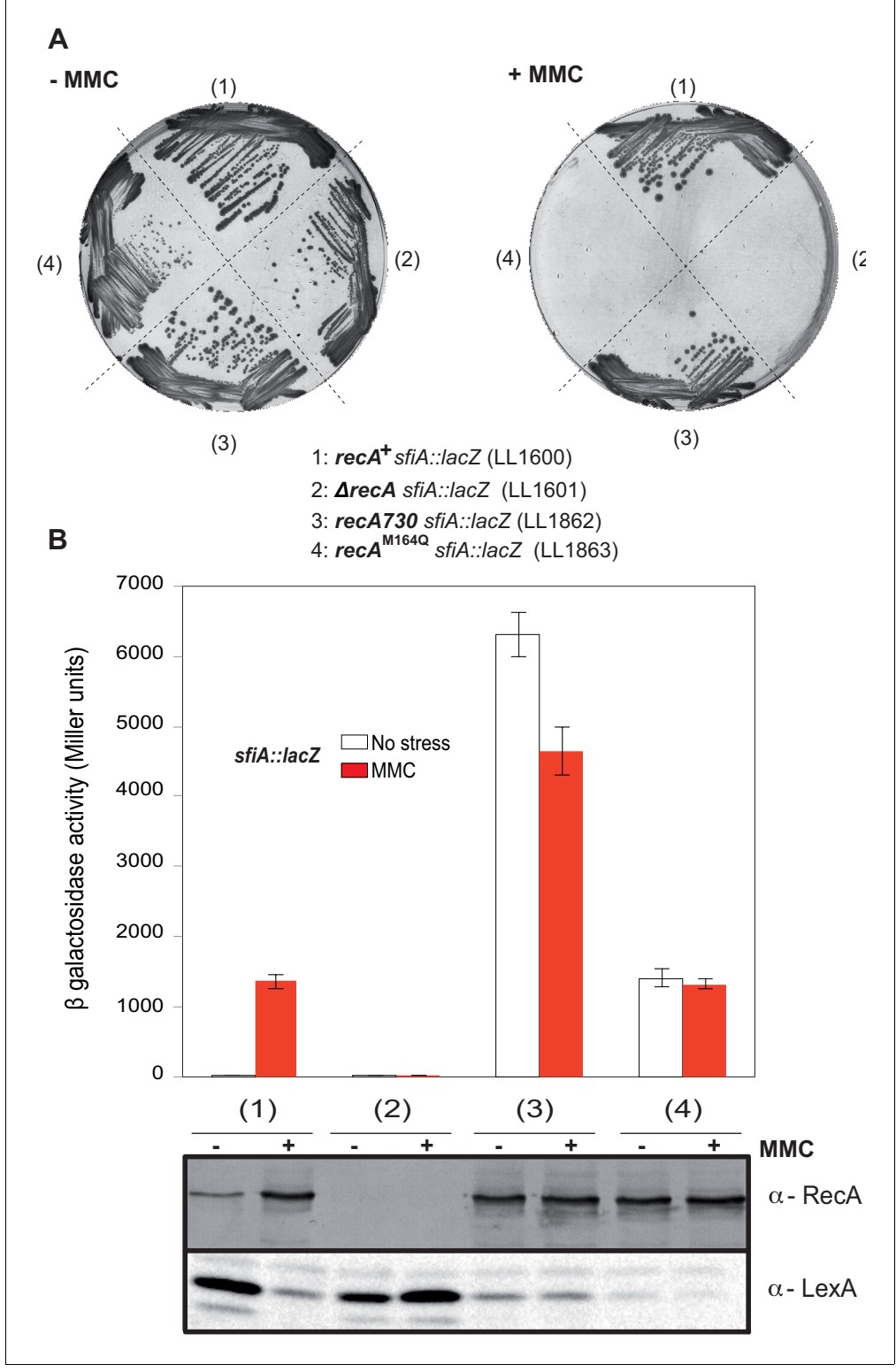

**Figure 6.** Characterization of the *recA*-M164Q chromosomal allele and comparison with *recA730*. The *sfiA::lacZ* fusion was inserted in wild type (WT) (LL1600), Δ*recA* (LL1601), *recA730* (LL1862), and *recA*-M164Q (LL1863) strains. (**A**) Strains tested, as indicated under the figure, were incubated on LB plates in the presence or absence of mitomycin C (MMC). (**B**) SOS induction after 2 hr exposure to MMC was monitored by assaying *sfiA::lacZ* expression (top panel) and western blot anti-RecA and anti-LexA analyses (bottom panel).

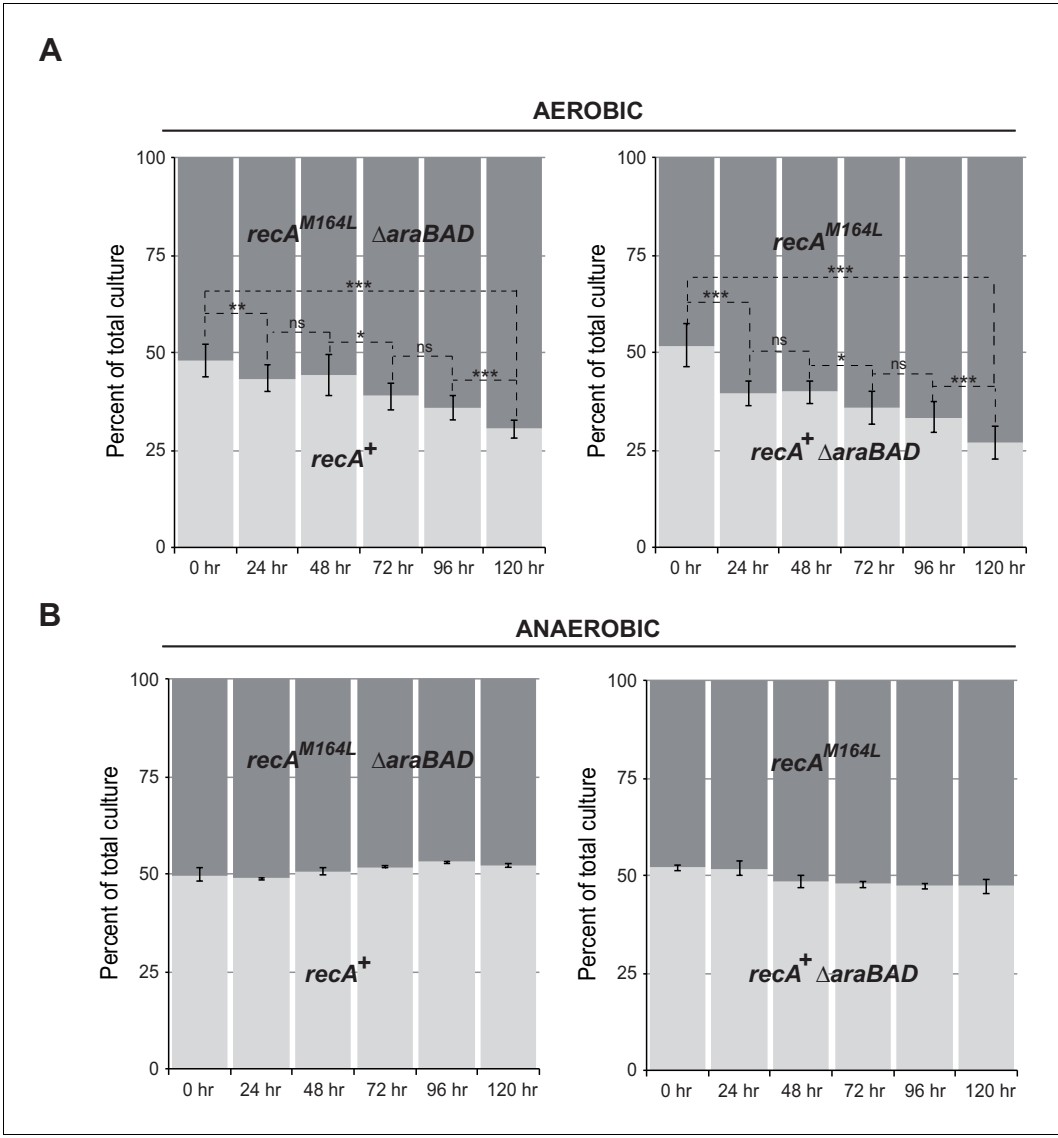

**Figure 7.** *recA*-M164L chromosomal allele enhances growth competition of *E. coli* in mixed aerobic culture. (**A**) In aerobic condition, *recA*-M164L outcompetes wild type (WT). *E. coli recA*WT (MG1655) and *recA*-M164L (LL1781) mutant strains were grown overnight in (LB)-rich medium. Cultures were then diluted into fresh LB medium and used in co-cultures with *recA*WT and *recA*-M164L Δ*araBAD* (LL1976) or *recA*WT Δ*araBAD* (LL1974) and *recA*-M164L (LL1781). Δ*araBAD* is a neutral mutation used to differentiate strains in the competition assay; loss of the *araBAD* operon results in red colonies on tetrazolium arabinose medium. The cell density was adjusted to obtain a 1:1 ratio of each of the two strains. The two strains were then grown together in fresh LB medium over 24 hr periods. Total cell titers were determined by serial dilutions on LB-tetrazolium arabinose agar plates after 0, 24, 48, 72, 96, and 120 hr of culture. (**B**) In anaerobic condition, a 1:1 ratio of *recA*-M164L and *recA*WT remains over time. The same protocol was used under anaerobic condition with LB containing 0.2% glucose. Asterisks indicate a statistically significant difference, *$p \leq 0.05$; **$p \leq 0.01$, and ***$p \leq 0.001$, and ns indicates a not statistically significant difference (Mann–Whitney U test).

in recombination-mediated repair. Then, we asked whether making that Met164 position refractory to oxidation would be beneficial or detrimental. For this, we introduced the *recA*-M164L allele in the chromosome and ran a growth competition assay between the WT and *recA*-M164L-containing strains. Under aerobic condition, the *recA*-M164L containing strain showed a clear advantage over the WT strain as, after a 120 hr growth, the population of the mutant counted for 70% of the total (*Figure 7A, B*). In contrast, under anaerobic condition, both strains grew at the same rate and a

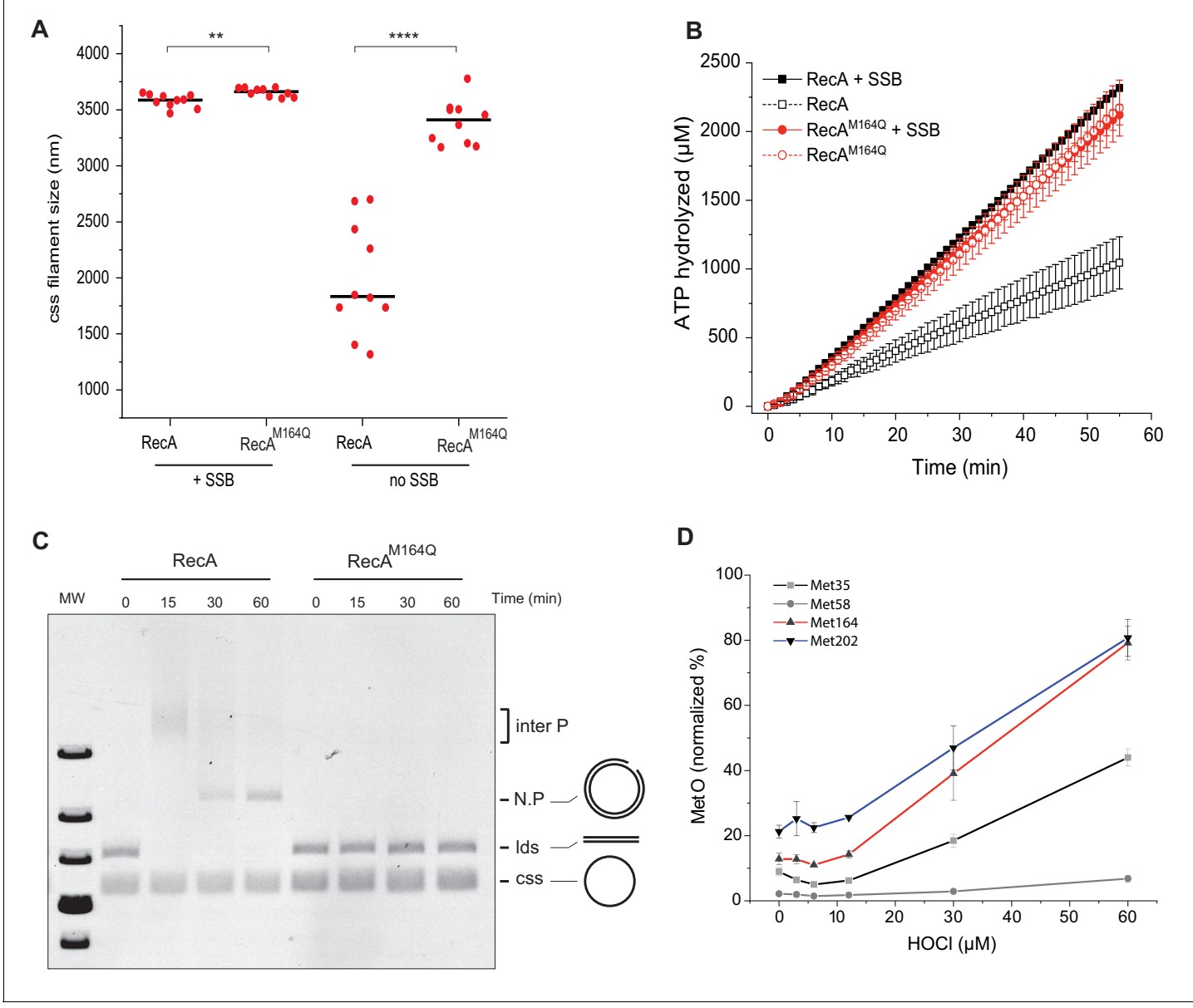

**Figure 8.** Biochemistry characterization of the RecA$^{M164Q}$, a position highly prone to oxidation. (**A**) Size measurement of cssDNA-RecA and cssDNA-RecA$^{M164Q}$ filament observed by electron microscopy in the absence or presence of single-strand binding protein (SSB). Red dots indicate the size of each individual filament, and black lines indicate the median values. Asterisks indicate a statistically significant difference, **p≤0.01 and ****p≤0.0001 (Mann–Whitney) ; n = 10. (**B**) ATP hydrolytic activity of RecA and RecA$^{M164Q}$ in the presence or absence of SSB. (**C**) RecA$^{M164Q}$ is unable to invade and/or catalyze DNA strand exchange. (**D**) Purified RecA protein was treated with increasing concentration of HOCl followed by mass spectrometry analysis. The relative percentage of Met-O was determined. Met residues showed intrinsic sensitivity to HOCl depending upon their position in the polypeptide. Error bar ± means s.d., n = 3. css: circular single strand.

The online version of this article includes the following figure supplement(s) for figure 8:

**Figure supplement 1.** RecA$^{M164Q}$ double-strand affinity is reduced.

steady 50% ratio of both population was kept throughout the growth (*Figure 7A, B*). This experiment showed that preventing oxidation of Met164 position provides *E. coli* with a better fitness.

Taken together, these results strongly show that oxidation of Met164 modifies both repair and regulatory properties of RecA. Unexpectedly, oxidation of Met164 constitutively activates the SOS system, yet the consequences appear to be physiologically deleterious for *E. coli*.

## Biochemical characterization of the RecA$^{M164Q}$ protein

We conducted biochemistry-based studies to investigate the mechanistic basis of the SOS-constitutive and repair-deficient phenotype conferred by the RecA$^{M164Q}$ variant. We purified RecA$^{M164Q}$ and added it along with SSB protein to cssDNA (*Lusetti et al., 2003b*; *Weinstock et al., 1981*; *Cox and Lehman, 1982*; *Kowalczykowski and Krupp, 1987*; *Lindsley and Cox, 1990*). We observed the formation of extended nucleoprotein filaments of similar or identical size as those obtained with WT RecA (*Figure 8A*). Surprisingly, SSB was not required. In the absence of SSB, RecA$^{M164Q}$ produced normal-length nucleoprotein filaments, whereas RecA$^{wt}$ formed only short filaments (*Figure 8A*). Next, enzymatic assays revealed that SSB was dispensable to facilitate ATPase activity of the RecA$^{M164Q}$ variant in the presence of cssDNA from M13 phage, and the apparent kcat of RecA$^{M164Q}$ was 21.75 $\mu$M·min$^{-1}$ compared with the 10.78 $\mu$M·min$^{-1}$ observed for the RecA$^{wt}$ (*Figure 8B*). Because at such concentration SSB is known to stimulate RecA-promoted nucleoprotein filament formation by melting DNA secondary structures (*Kowalczykowski and Krupp, 1987*), these observations indicated that the RecA$^{M164Q}$ variant displayed an enhanced filament propagation capacity, pointing to a potential hyper-reactivity of RecA toward ssDNA. Electrophoresis mobility shift assay (EMSA) techniques showed that the RecA$^{M164Q}$ variant binds cssDNA with the same affinity as RecA but fails to bind circular double strand (cds) nicked DNA (*Figure 8—figure supplement 1*). These observations provided a molecular explanation as to why changing Met164 for Q, by inference oxidizing Met164 in Met-O, enabled RecA to turn on the SOS response even in the absence of MMC-induced DNA damage. We observed that the RecA$^{M164Q}$ variant was unable to initiate the DNA strand exchange, suggesting a defect in homologous pairing (*Figure 8—figure supplement 1*). The failure of the variant to promote DNA strand exchange *in vitro* accounts for the *in vivo* hypersensitivity to MMC exposure of a strain synthesizing the RecA$^{M164Q}$ variant. Altogether, these biochemical investigations provided a detailed analysis of features that RecA lost upon oxidation of Met164 and provided us with a molecular explanation for both *in vivo* constitutive SOS activation and failure to repair MMC-inflicted damages.

## M164 a residue highly prone to oxidation

The results above revealed a high level of heterogeneity between sensitivity of the different Met residues to oxidative agents. Besides the fact that the consequences of oxidation varied depending upon which Met residue had been modified, Met residues also showed various intrinsic sensitivities to HOCl depending upon their position in the polypeptide. To complete our study, we investigated whether we could further classify the Met residue as a function of their sensitivity to increasing concentration of HOCl. We focused our attention on the four Met residues studied above, that is, Met35, 58, 164, and 202. Met58 proved to be totally resistant to oxidation (*Figure 8D*). This is consistent with the fact it is not solvent-exposed. Met202 was the more sensitive. Interestingly, mimetic oxidation variant RecA$^{M202Q}$ has been previously studied (*McGrew and Knight, 2003*; *Hörtnagel et al., 1999*) and shown to exhibit properties close to WT, strongly supporting the notion that oxidation at that position would bear no effect on RecA activity. Met164 was extremely sensitive as well, although slightly less reactive than Met202, but far more sensitive than Met35 (*Figure 8D*). These results showed that Met residues are not equivalent in the face of a potent oxidative agent such as HOCl and that M164 is highly prone to oxidation.

## Discussion

ROS-induced DNA lesions have been studied for many decades, particularly in relation to their effect on mutagenesis and cancer. The present work provides a new insight into the ROS-induced challenge to DNA integrity by showing that RecA, a key element in recombinational repair and SOS induction, is itself under ROS threat, uncovering a new layer in the relationship between oxidative stress and DNA repair/recombination as hampering RecA activity by oxidation will bear wide consequences in many DNA-associated transactions. The same is very much likely to hold in more complex organisms such as humans as RecA is highly conserved and consequences of human RecA oxidation in DNA damage-associated pathologies are to be expected as well.

In this study, we collected multiple evidences showing that RecA is targeted by ROS and that MsrA/B maintains a level of reduced functional RecA necessary to carry out both efficient

recombination and SOS regulation. Importantly, we cannot rule out the possibility that other recombination/DNA repair-associated proteins, like RecBCD and LexA of *E. coli,* could be also targeted by ROS. As a matter of fact, a few previous studies had pointed out a link between oxidative stress and activity of proteins involved in genome integrity. In *Neisseria*, a LexA-like protein was found to be inactivated by ROS, thereby allowing full expression of a SOS-like regulon and enhanced DNA repair (*Schook et al., 2011*). In *Deinococcus radiodurans*, resistance to DNA damage caused by radiation was proposed to be mostly dependent on damage-resistant DNA repair proteins (*Daly et al., 2007*; *Krisko and Radman, 2013*). Also, a previous study reported oxidation of RPA, the eukaryotic SSB counterpart. RPA protein pondered its potential consequences for skin cancer risk due to a defect in nucleotide excision repair (*Guven et al., 2015*). However, in all the cases cited above, oxidation was irreversible. In contrast, here we show that oxidation of RecA can be reversed by the action of methionine sulfoxide reductases MsrA/B. This provides a very different view of oxidation as reversibility afforded by MsrA/B opens the possibility to exploit redox changes to regulate RecA protein activities. A similar situation arises with HypT, a transcriptional regulator relying on Met residues to sense HOCl-mediated oxidation (*Drazic et al., 2013*). Conceptually, the same reversibility phenomenon arises with Cys positions whose oxido-reduction can be controlled by thioredoxin/glutaredoxin systems. Altogether, these different cases recall that oxidation is not only a damaging event but also a signal the cells can exploit to set up adaptive responses.

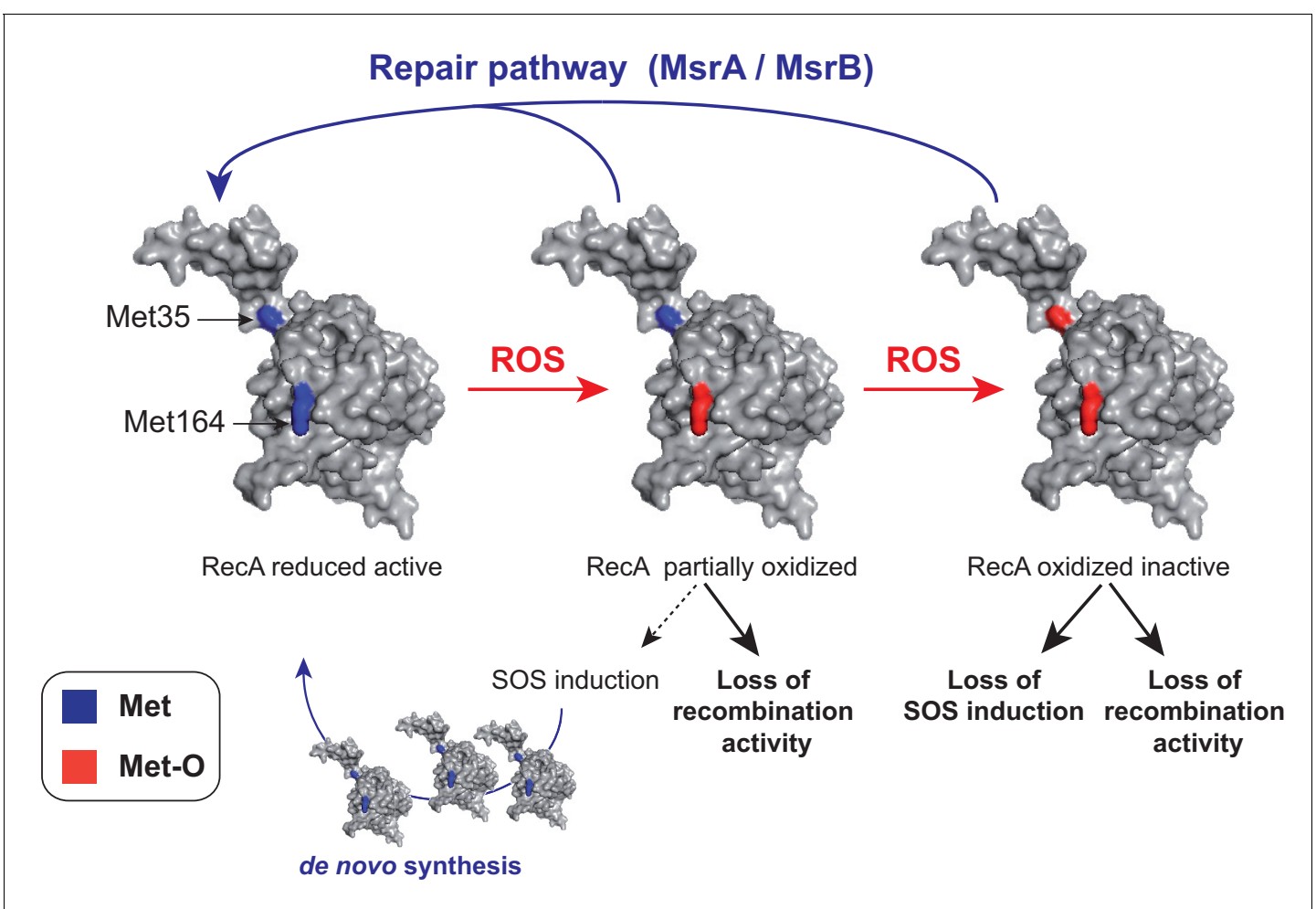

**Figure 9.** A model of homeostasis control of RecA under oxidative stress. RecA structural model is shown with the two Met residues (Met35 and Met164) analyzed in the present work. The RecA partially oxidized form contains a Met164-oxidized residue while the RecA-oxidized inactive form contains both Met35- and Met164-oxidized residues. The MsrA/B repair pathway allows to maintain the pool of functional RecA by repair, and the SOS induction pathway allows to replenish the pool of functional RecA by *de novo* synthesis.

Genetic and biochemical analyses as well as use of Gln as a proxy for Met oxidation led us to observe a variety of functional consequences of oxidation on RecA activity. Exposure of RecA to $H_2O_2$ or HOCl led to oxidation of Met35, 58, 164, and 202, and loss of all biochemical activities. Moreover, dose–response analysis revealed that Met residues responded with a different sensitivity to ROS as some Met residues did not undergo oxidation while others did. This illustrated how oxidation of a protein covers a complex and heterogenous situation. Hence, the present study demonstrates that in order to appreciate and predict the effects of Met oxidation of a given protein activity, the following information will be needed: (i) functional consequences of oxidation of each Met, (ii) proportion of Met-O-containing polypeptides within the whole oxidized population, (iii) efficiency of Msr repair acting at each Met position, and (iv) threshold value of activity of the targeted protein for a WT phenotype be maintained.

Oxidation of two Met residues, Met35 and Met164, and its associated functional consequences was particularly studied. Mimicking oxidation of Met35 alone also led to total loss of function. This endows Met35 with a key role in RecA activity, somehow at odds with the fact that it is not conserved throughout the RecA/Rad51 family (*Appendix 1—figure 1*; *Lusetti and Cox, 2002*). Oxidation of Met164 led to loss of recombination capacity as indicated by sensitivity to MMC treatment, decreased fitness, and an incapacity to promote DNA strand exchange *in vitro*. Also, Met164 oxidation caused constitutive activation of SOS response *in vivo*. However, growth characteristic revealed that this was of no advantage for *E. coli,* and reciprocally, a mutation that prevented oxidation at that position provided a better fitness over the WT. Interestingly, Met164 is conserved in the human and *Saccharomyces cerevisiae* Rad51 proteins (*Appendix 1—figure 1*). Hence, the overall picture that emerged is that oxidation of RecA per se is deleterious for the cell and the MsrA/B repair system helps to mitigate this effect.

Analysis of oxidation of Met35 and Met164 and of its functional consequences provides us with a model on the pathway followed by RecA during oxidation and how cell can replenish the intracellular pool of functional RecA under oxidative stress (*Figure 9*). One mode is to exploit the MsrA/B system to repair existing and oxidized RecA. This is the situation that will prevail when MsrA/B acts at the Met35 position. Another mode is to start off synthesis of fresh RecA protein by activation of the SOS regulon, which includes the *recA* gene. This is the possibility afforded by the oxidation of Met164 (*Figure 9*). A mechanistic question still remaining is how oxidation of Met164 facilitates SOS activation in the absence of free 3′ DNA ends. Our observation that RecA$^{M164Q}$ does not seem to require the help of SSB for nucleofilament formation and ATP hydrolysis opens the possibility that, like RecA730 (*Long et al., 2008*), the RecA$^{M164Q}$ variant might be activated by single-stranded DNA produced upon replication. Another possibility is that homologous recombination deficiency of the RecA$^{M164Q}$ variant activates constitutively the SOS because of accumulation of DNA damages.

To conclude, this study identified RecA as a target of oxidative stress. RecA-associated recombination/repair and SOS-inducing properties are potentially canceled following exposure to oxidative stress and cells make use of MsrA/B as a protecting/repairing system, which suppresses most oxidative-mediated defects. Thus, RecA stands as a new bona fide substrate of the MsrA/B anti-ROS defense system.

## Materials and methods

### Key resources table

| Reagent type (species) or resource | Designation | Source or reference | Identifiers | Additional information |
|---|---|---|---|---|
| Strain, strain background (*Escherichia coli*) | MG1655 | LCB Collection | N/A | |
| Strain, strain background (*Escherichia coli*) | BE152 | This study | N/A | ∆msrA ∆msrB |

*Continued on next page*

Continued

| Reagent type (species) or resource | Designation | Source or reference | Identifiers | Additional information |
|---|---|---|---|---|
| Strain, strain background (*Escherichia coli*) | CH011 | This study | N/A | Δ*msrA*::FRT Δ*msrB*::FRT |
| Strain, strain background (*Escherichia coli*) | CH074 | This study | N/A | Δ*recA*::kan |
| Strain, strain background (*Escherichia coli*) | LL401 | *Loiseau et al., 2007* | N/A | *ara$_P$*::*erpA* |
| Strain, strain background (*Escherichia coli*) | LL1594 | This study | N/A | Δ*recA*::spc |
| Strain, strain background (*Escherichia coli*) | BE007 | *Ezraty et al., 2014* | N/A | Hpx⁻ = Δ*katE* Δ*katG ahpC*::kan |
| Strain, strain background (*Escherichia coli*) | BE080 | This study | N/A | BE007 Δ*msrA* Δ*msrB* |
| Strain, strain background (*Escherichia coli*) | LL1609 | This study | N/A | BE007 Δ*recA*::spec |
| Strain, strain background (*Escherichia coli*) | BE032 | This study | N/A | BE007 Δ*uvrA* |
| Strain, strain background (*Escherichia coli*) | BE033 | This study | N/A | BE007 Δ*uvrA* Δ*msrA* Δ*msrB* |
| Strain, strain background (*Escherichia coli*) | *sfiA*::*lacZ* | *Grompone et al., 2004* | N/A | Δ*lacZ*::cat *sfiA*::Mud *lacZ* cat-bla |
| Strain, strain background (*Escherichia coli*) | CH039 | This study | N/A | CH011 Δ*lacZ*::cat *sfiA*::Mud *lacZ* cat-bla |
| Strain, strain background (*Escherichia coli*) | LL1713 | This study | N/A | *recA*::gfp-cat |
| Strain, strain background (*Escherichia coli*) | EAW831 | This study | N/A | *sfiA* ::FRT *recA*$^{M164Q}$ |
| Strain, strain background (*Escherichia coli*) | EAW287 | *Robinson et al., 2015* | N/A | *sfiA* ::FRT *recA*$^{E38K}$::kan (*recA730*) |
| Strain, strain background (*Escherichia coli*) | LL1601 | This study | N/A | MG1655 Δ*lacZ*::cat *sfiA*::Mud *lacZ* cat-bla Δ*recA*::spc |
| Strain, strain background (*Escherichia coli*) | LL1862 | This study | N/A | MG1655 Δ*lacZ*::cat *sfiA*::Mud *lacZ* cat-bla *recA*$^{E38K}$::kan (*recA730*) |
| Strain, strain background (*Escherichia coli*) | LL1863 | This study | N/A | MG1655 Δ*lacZ*::cat *sfiA*::Mud *lacZ* cat-bla + *recA*$^{M164Q}$ |
| Strain, strain background (*Escherichia coli*) | LL1976 | This study | N/A | MG1655 *recA*$^{M164L}$ Δ*araBAD*::kan |

*Continued on next page*

Continued

| Reagent type (species) or resource | Designation | Source or reference | Identifiers | Additional information |
|---|---|---|---|---|
| Strain, strain background (*Escherichia coli*) | LL1781 | This study | N/A | MG1655 *recA*^M164L |
| Strain, strain background (*Escherichia coli*) | LL1974 | This study | N/A | MG1655 Δ*araBAD::kan* |
| Strain, strain background (*Escherichia coli*) | BW25113 | LCB Collection | N/A | F⁻, Δ(*araD-araB*)567, *lacZ*4787 (del)::*rrnB*-3, LAM⁻, *rph*-1, Δ(*rhaD-rhaB*)568, *hsdR*514 |
| Strain, strain background (*Escherichia coli*) | STL2669 | A gift from S Lovett | N/A | (Δ*recA-srlR*)306::Tn*10* Tet^R*xonA2* (exoI⁻) |
| Strain, strain background (*Escherichia coli*) | DH5α | LCB Collection | N/A | F⁻ *endA1 glnV44 thi-1 recA1 relA1 gyrA96 deoR nupG purB20* φ80d*lacZ*ΔM15 Δ(*lacZYA-argF*)U169, hsdR17(r_K⁻m_K⁺), λ⁻ |
| Plasmids | pGEMT-easy | Promega | A1360 | 'High' copy vector, cloning site into *lacZ* ApR |
| Plasmids | pUC18 | LCB Collection | N/A | High copy vector with the small size MCS inverted as pUC19 ApR |
| Plasmids | pMsrA | LCB Collection | N/A | pUC18 - *msrA*⁺ |
| Plasmids | pMsrB | LCB Collection | N/A | pUC18 - *msrB*⁺ |
| Plasmids | pRecA | A gift from M. Modesti | N/A | pUC18 - *recA*⁺ |
| Plasmids | pBBR | LCB Collection | N/A | 'Medium-low' copy vector pBBR-MCS2 + cat HindIII CmR KanR |
| Plasmids | pBBR-RecA | This study | N/A | pBBR - *recA* SacII/SpeI |
| Plasmids | pBBR-RecAM35L | This study | N/A | pBBR - *recAM35L* SacII/SpeI |
| Plasmids | pBBR-RecAM164L | This study | N/A | pBBR - *recAM164L* SacII/SpeI |
| Plasmids | pBBR-RecAM35Q | This study | N/A | pBBR - *recAM35Q* SacII/SpeI |
| Plasmids | pBBR-RecAM164Q | This study | N/A | pBBR - *recAM164Q EcoRI* |
| Plasmids | pEAW1078 | This study | N/A | Derived from pEAW260 - *recAM35L* ApR |
| Plasmids | pCJH0002 | This study | N/A | Derived from pEAW260 - *recAM164Q* ApR |
| Plasmids | pT7pol26 | Cox Lab Collection | N/A | Carries the T7 RNA polymerase under the control of a *lac* promoter |
| Sequenced-based reagent | recA for | This study | PCR primers | CGGTGCGTCGTCAGGCTACTGCGT |
| Sequenced-based reagent | recA rev | This study | PCR primers | GCCAGAATGCGTACCGCACGAT |
| Sequenced-based reagent | M35L for | This study | PCR primers | GACCGTTCCCTGGATGTGGAAA CCATCTCTACCGGTT CGCTTTCACTGGATATCGCGCTT |
| Sequenced-based reagent | M35L rev | This study | PCR primers | GGTTTCCACATCCA GGGAACGGTCTTCA CCCAGGCGCATG |
| Sequenced-based reagent | M35Q for | This study | PCR primers | GACCGTTCCCAGGATGTG GAAACCATCTCTACCGG TTCGCTTTCACTGGATATCGCG |

*Continued on next page*

*Continued*

| Reagent type (species) or resource | Designation | Source or reference | Identifiers | Additional information |
|---|---|---|---|---|
| Sequenced-based reagent | M35Q rev | This study | PCR primers | TTCCACATCCTGGGAACGGTC TTCACCCAGGCGCAT GATGGAGCC |
| Sequenced-based reagent | M164L for | This study | PCR primers | GGCGACTCTCACCTGGGCC TTGCGGCACGTATGATGAGCCAGGCG |
| Sequenced-based reagent | M164L rev | This study | PCR primers | TGCCGCAAGGCCCAGGTGAGAGTCG CCGATTTCGCCTTCGATTTC |
| Sequenced-based reagent | M164Q for | This study | PCR primers | GGCGACTCTCACCAGGGCCTTG CGGCACGTATGATGAGCCAGGCG |
| Sequenced-based reagent | M164Q rev | This study | PCR primers | TGCCGCAAGGCCCTGGTGAGAGTC GCCGATTTCGCCTTCGATTTC |
| Sequenced-based reagent | T7 pro | This study | PCR primers | TAATACGACTCACTATAGGG |
| Sequenced-based reagent | recA*-M35L-rev | This study | PCR primers | GTAGAGATGGTTT CCACATCCAGGGAA CGGTCTTC |
| Sequenced-based reagent | recA*-M164Q-rev | This study | PCR primers | TGCCGCAAGGCCCT GGTGAGAGTCGCCG |
| Sequenced-based reagent | 5for recA M165L SceI K7kan | This study | PCR primers | GGCGACTCTCACCTGGG CCTTGCG GCACGTATG ATGAGCCAGGC GTGTAGGCTGGAGCTGCTTC |
| Sequenced-based reagent | 3rev recA K7Kan | This study | PCR primers | AATTTTCATACGGATCTGGTTGAT GAAGATCAGCAGCGTCCT ATTACCCTGTTATCCCTACC TTAGTTCCTATTCCGAAGTTC |
| Sequenced-based reagent | 5-*recA*-GFP | This study | PCR primers | GATGATAGCGAAGGCGTA GCAGAAACTAACGAAGA TTTTTAATAAGAAGGAG ATATACATATGAG |
| Sequenced-based reagent | 3-*recA*-K7Cat | This study | PCR primers | AGGGCCGCAGATGCGACCCT TGTGTATCAAACAAGA CGATTATCACTTATTCAGGCGTA |
| Sequenced-based reagent | 5-K7wanner-spc | This study | PCR primers | TGTGTAGGCTGGAGCTGCTTC GAAGTTCCTATACTT TCTAGCAGGAGGAATTCACCAT GAGTGAAAAAGTGCCCGCC |
| Sequenced-based reagent | 3-K7wanner-spc | This study | PCR primers | CATATGAATATCCTCCTT AGTTCCTATTCCGAAGT TCCTATTCTCTAGAAAG TATCATTGGCTGGCA CCAAGCAGTTTA |
| Sequenced-based reagent | 5-wanner-recA | This study | PCR primers | ATTGACTATCCGGTATTACCCGGCA TGACAGGAGTAAAAGTGTAGGCTG GAGCTGCTTC |
| Sequenced-based reagent | 3-wanner-recA | This study | PCR primers | AGGGCCGCAGATGCGACCCTTGT GTATCAAACAAGACGACATATG AATATCCTCCTTA |
| Antibody | Anti-RecA (Rabbit polyclonal) | Abcam | 63797 | WB (1:20,000) |
| Antibody | Anti-LexA (Rabbit polyclonal) | P. Moreau (LCB) | | WB (1:5,000) |
| Antibody | Goat anti-rabbit IgG-Alk Phos (polyclonal) | Chemicon International | AP132A | WB (1 :10,000) |
| Antibody | Goat anti-rabbit HRP (polyclonal) | Promega | W4018 | WB (1 :10,000) |

*Continued on next page*

*Continued*

| Reagent type (species) or resource | Designation | Source or reference | Identifiers | Additional information |
|---|---|---|---|---|
| Commercial assay or kit | ECL western blotting kit | Thermo Scientific | 32106 | |
| Commercial assay or kit | Gel purified using kit | Promega | A9282 | |
| Commercial assay or kit | Protein assay kit | Bio-Rad | 23236 | |
| Software, algorithm | QI-Macros software | | KnowWare International, Inc, CO | |
| Chemical compound, drug | ProteaseMAX | Promega | V2071 | |
| Chemical compound, drug | Trypsin Gold | Promega | V5280 | |
| Chemical compound, drug | Sodium hypochlorite | Acros Organics | AC219255000 | |
| Chemical compound, drug | Phusion enzyme | Thermo Scientific | F531S | |
| Chemical compound, drug | LB (Lysogeny Broth) | Difco | 244620 | |
| Other | Vinyl Anaerobic Chambers | Coy | Coy Laboratory Products Inc, USA | coylab.com |
| Other | Typhoon FLA-9000 | GE Healthcare | FLA9000 | |
| Other | | Thermo Scientific | F531S | |

## Strains

All strains are listed in the Key resources table. All strains used for *in vivo* experiments are *E. coli* MG1655 derivatives. Strains were constructed by transduction from P1 grown on strains from Keio or LCB collections (*Baba et al., 2006*), or they were constructed as described by *Datsenko and Wanner, 2000* protocol using lambda Red recombination. pCP20 was used to remove the cassette when it was required. All constructions were confirmed by PCR. For strains carrying spectinomycin (spec) resistance cassette, a derivative Datsenko–Wanner protocol was used. Briefly, an FRT-flanked spectinomycin resistance (*spc*) cassette was amplified using chromosomal DNA from LL401 (*ara$_p$::erpA*) strain as a template, 5-K7wanner-spc forward primer, and 3-K7wanner-spc reverse primer (Key Resources Table). The amplified *spc* cassette fragment was inserted into pGEMTeasy vector by TA cloning, yielding pGEMT-spc. *recA::spc* strain was then constructed in a one-step inactivation, and a DNA fragment containing *spc* gene flanked with a 5′ and 3′ region bordering *recA* gene was amplified by PCR using pGEMT-spc cassette as a template and oligonucleotides 5wanner-recA and 3wanner-recA (Key Resources Table). Strain BW25113, carrying the pKD46 plasmid, was transformed by electroporation with the amplified fragment, and Spc-resistant colonies were selected. The replacement of chromosomal *recA* gene by *spc* gene was checked by PCR amplification. Phage P1 was used to transduce the mutation into MG1655 and Hpx⁻ strains, yielding *recA::spc* and Hpx⁻ *recA::spc* strains, respectively.

The '*recA–gfp*' fusion *gfp* gene was introduced just after the *recA* gene without removing the stop codon of *recA* in the chromosome. The *gfp* gene has its own ribosome binding site, and the *recA–gfp* fusion is transcriptional. For construction of the single-copy *recA-gfp*+ fusion, a fragment containing the promoterless *gfp*+ gene and the chloramphenicol resistance cassette was amplified from the P*prgH* plasmid with primers 5-recA-GFP and 3-recA-K7Cat (Key Resources Table). All gfp+ constructs were first integrated into the chromosome of *E. coli* strain BW25113 by using the Lambda Red system. Phage P1 was then used to transduce the *recA-gfp*+ fusion into MG1655, Hpx⁻, Δ*msrA* Δ*msrB*, and Hpx Δ*msrA* Δ*msrB* strains.

## Plasmids

All plasmids used were sequenced to confirm their construction and are listed in the Key Resources Table. Plasmids with the *recA* wt or *recA* with methionine mutations were constructed by the two-step PCR fragments method using oligonucleotides listed in the Key Resources Table, constructs were sub-cloned in pGEMT-easy vector (Promega), and then plasmids were double digested by SacII/SpeI and cloned in pBBR-MC2LL*. Plasmids used for the suppression and complementation of *msrA msrB* deletions phenotypes are derivatives of pUC18 vector. Plasmid used for RecA^M35L and RecA^M164Q overexpression was constructed by the direct mutagenesis of the pEAW260 using modified Megaprimer protocol (*Kirsch and Joly, 1998*). The PCR program was adapted for the Phusion polymerase enzyme. In both cases, T7-pro was used as the upstream primer and the downstream primer was recA*-M35L-rev or recA*-M164Q-rev.

## Reagents and antibiotics

All chemicals used were purchased from Sigma, Euromedex, or Fisher Scientific. Antibiotics were used at the following concentrations: chloramphenicol (25 µg/mL), ampicillin (50 µg/mL), kanamycin (50 µg/mL), and spectinomycin (75 µg/mL).

## Growth conditions

Cells were grown in Lysogeny Broth (LB) medium; for solid medium, 15 g/L of agar was supplemented. In aerobic condition, cells were grown in flask with a 1/10 vol of the flask capacity at 37°C with shaking at 150 rpm. In anaerobic condition, cells were grown in Coy Chamber; the medium was preincubated at least 16 hr before the experiments.

## DNA substrates

The M13mp18 RFI and css forms were prepared as described (*Haruta et al., 2003*) or purchased from Biolabs. The M13mp18 lds was obtained by the digestion of the M13mp18 RFI with the *Bam*HI enzyme or by the double digestion with the *Bgl*II and *Dra*III enzymes (use only for the EM experiment). The cds nicked DNA was obtained by the digestion of pEAW951 (pBR22 carrying 2000 bp of M13mp18) with the DNaseI following the protocol described by *Shibata et al., 1981*. Digest products were gel purified using Promega kit. All DNA concentrations are reported in terms of µM of nucleotides (µM/nt).

## Proteins

The *E. coli* MsrA and MsrB (*Grimaud et al., 2001*), RecA (*Cox et al., 1981*; *Britt et al., 2011*), and SSB (*Lohman et al., 1986*) proteins were purified as described previously. RecA^M35L and RecA^M164Q were purified following the same protocol as RecA wt (*Cox et al., 1981*; *Britt et al., 2011*). The concentrations of MsrA and MsrB were determined by the Bradford method using the Bio-Rad Protein assay kit. The concentrations of SSB and RecA proteins were determined using native extinction coefficients: $\varepsilon = 2.23 \times 10^4$ M$^{-1}$ cm$^{-1}$ for RecA proteins and $\varepsilon = 2.38 \times 10^4$ M$^{-1}$ cm$^{-1}$ for SSB. All proteins were tested for endo- and exonuclease contamination. DNA degradation of cds, lds, and cssDNA by the proteins prep was assayed. No contaminating endo- or exonuclease activity was detected.

## *In vivo* analysis of RecA by gel shift assays

WT and Δ*msrA* Δ*msrB* mutants were grown aerobically at 37°C with shaking (150 rpm) in 10 mL of LB medium in 100-mL flasks. At $A_{600nm} \approx 0.5$, cells were subjected to NaOCl treatment (1.75 or 2 mM). Cells were then incubated at 37°C with shaking for 2 hr. Samples were precipitated with trichloroacetic acid (TCA), and the pellets were then washed with ice-cold acetone, suspended in SB buffer, heated at 95°C, and loaded on a 12% SDS-PAGE gel for immunoblot analysis using anti-RecA antibodies (Abcam 63797). The protein amounts loaded were standardized by taking into account the $A_{600nm}$ values of the cultures.

## RecA oxidation and reparation assay

RecA oxidation was performed in 1× RecA buffer (25 mM Tris-acetate, 10 mM magnesium acetate, 3 mM potassium glutamate and 5% [w/v] glycerol), 50 mM hydrogen peroxide (H$_2$O$_2$) was added to

1.05 µM RecA, and the mixture was incubated for 2 hr at 37°C. Oxidation reaction was stopped by adding 400 U/mL catalase bovine, and the mix was incubated for 45 min at room temperature to remove the $H_2O_2$. The non-oxidized RecA was treated similarly except that nuclease-free water was used instead of $H_2O_2$. MsrA/B reparation assay was performed using aliquot of RecA oxidized, 2 µM MsrA, 10 µM MsrB, and 50 mM DTT (used for the enzyme recycling activity of Msr) were added, and the reaction was incubated 2 hr at 37°C to allow the reparation. In control, non-oxidized RecA was treated similarly except that RecA conservation buffer was used instead of MsrA and MsrB. Also, 2 µL of each reaction was mixed with 3 µL of cracking buffer heated for 10 min and run on 12% Tris glycine SDS-PAGE. Gels were stained with Coomassie Blue, the protein bands were cut and analyzed in-gel digestion, and mass spectrometric analysis was done at the Mass Spectrometry Facility (Biotechnology Center, University of Wisconsin-Madison).

### In-gel digestion (*Figure 3A*)

The digestion was performed as outlined on the website: http://www.biotech.wisc.edu/ServicesResearch/MassSpec/ingel.htm. In short, Coomassie Blue R-250 stained gel pieces were de-stained twice for 5 min in MeOH/$H_2O$/$NH_4HCO_3$(50%:50%:100 mM), dehydrated for 5 min in ACN/$H_2O$/$NH_4HCO_3$(50%:50%:25 mM), then once more for 1 min in 100% ACN, dried in a Speed-Vac for 2 min, reduced in 25 mM DTT (dithiotreitol in 25 mM $NH_4HCO_3$) for 30 min at 52°C, alkylated with 55 mM IAA (iodoacetamide in 25 mM $NH_4HCO_3$) in darkness at room temperature for 30 min, washed twice in $H_2O$ for 30 s, equilibrated in 25 mM $NH_4HCO_3$ for 1 min, dehydrated for 5 min in ACN/$H_2O$/$NH_4HCO_3$(50%:50%:25 mM), and then once more for 30 s in 100% ACN, dried again and rehydrated with 20 µL of trypsin solution (10 ng/µL trypsin Gold [Promega] in 25 mM $NH_4HCO_3$/0.01% ProteaseMAX w/v [Promega]). Additionally, 30 µL of digestion solution (25 mM $NH_4HCO_3$/0.01% ProteaseMAX w/v [Promega]) was added to facilitate complete rehydration and excess overlay needed for peptide extraction. The digestion was conducted for 3 hr at 42°C. Peptides generated from digestion were transferred to a new tube and acidified with 2.5% trifluoroacetic acid (TFA) to 0.3% final. Degraded ProteaseMAX was removed via centrifugation (max speed, 10 min) and the peptides solid phase extracted (*ZipTip C18* pipette tips Millipore, Billerica, MA).

### Mass spectrometry matrix-assisted laser desorption/ionization-time of flight-time of flight analysis (*Figure 3A*)

Peptides were eluted off the C18 SPE column with 1 µL of acetonitrile/$H_2O$/TFA (60%:40%:0.2%) into 1.5 mL Protein LoBind tube (Eppendorf), 0.5 µL was deposited onto the Opti-TOF 384-well plate (Applied Biosystems, Foster City, CA), and re-crystallized with 0.4 µL of matrix (5 mg/mL α-cyano-4hydroxycinnamic acid in acetonitrile/$H_2O$/TFA [60%:40%:0.1%]). Peptide Mass Fingerprint analysis was performed on a 4800 matrix-assisted laser desorption/ionization-time of flight-time of flight mass spectrometer (Applied Biosystems, Foster City, CA). In short, peptide fingerprint was generated scanning 700–4000 Da mass range using 1000 shots acquired from 20 randomized regions of the sample spot at 3600 intensity of OptiBeam on-axis Nd:YAG laser with 200 Hz firing rate and 3–7 ns pulse width in positive reflectron mode. Fifteen most abundant precursors, excluding trypsin autolysis peptides and sodium/potassium adducts, were selected for subsequent tandem MS analysis, where 1200 total shots were taken with 4200 laser intensity and 1.5 kV collision-induced activation using air. Post-source decay fragments from the precursors of interest were isolated by timed-ion selection and reaccelerated into the reflectron to generate the MS/MS spectrum. Raw data was deconvoluted using GPS Explorer software and submitted for peptide mapping and MS/MS ion search analysis with an in-house licensed Mascot search engine (Matrix Science, London, UK) with cysteine carbamidomethylation as fixed modification plus methionine oxidation and asparagine/glutamine deamidation as variable modifications. The relative Met-O % was calculated using the values of the isotope cluster area of the initial mass, the mass +16 and +32 Da for each peptide. The mean values of the biological triplicates are shown in the histogram plot.

### Analysis of Met residues sensitivity to HOCl by mass spectrometry (*Figure 8D*)

RecA oxidation was performed in buffer Tris-HCl 10 mM pH 7.4, DTT 0.1 mM, EDTA 0.01 mM, and glycerol 5%. RecA was incubated with varying molar excess of HOCl (Acros Organics) for 30 min

at 25°C. Following incubation, reaction was stopped with 50 mM of L-methionine. For the identification of peptides, samples were reduced and alkylated before digestion of O/N with trypsin at 30°C in 50 mM NH₄HCO₃(pH 8.0). Peptides were dissolved in solvent A (0.1% TFA in 2% ACN), directly loaded onto reversed-phase pre-column (Acclaim PepMap 100, Thermo Scientific), and eluted in backflush mode. Peptide separation was performed using a reversed-phase analytical column (Acclaim PepMap RSLC, 0.075 × 250 mm, Thermo Scientific) with a linear gradient of 4–36% solvent B (0.1% FA in 98% ACN) for 36 min, 40–99% solvent B for 10 min, and holding at 99% for the last 5 min at a constant flow rate of 300 nL/min on an EASY-nLC 1000 UPLC system. The peptides were analyzed by an Orbitrap Fusion Lumos tribrid mass spectrometer (Thermo Fisher Scientific). The peptides were subjected to NSI source followed by tandem mass spectrometry (MS/MS) in Fusion Lumos coupled online to the UPLC. Intact peptides were detected in the Orbitrap at a resolution of 120,000. Peptides were selected for MS/MS using HCD setting at 30; ion fragments were detected in the Orbitrap at a resolution of 30,000. A targeted mass list was included for the 11 theoretical peptide sequences (see *Table 1*), considering potential oxidation of Met and charges states +2 or +3. The electrospray voltage applied was 2.1 kV. MS1 spectra were obtained with an AGC target of 4E5 ions and a maximum injection time of 50 ms, and targeted MS2 spectra were acquired with an AGC target of 2E5 ions and a maximum injection time of 60 ms. For MS scans, the m/z scan range was 350–1800. The resulting MS/MS data was processed and quantified using Skyline (version 19.1.0.193). Trypsin was specified as cleavage enzyme allowing up to two missed cleavages, four modifications per peptide, and up to three charges. Mass error was set to 10 ppm for precursor ions and 0.05 Da for fragment ions. Oxidation on Met was considered as variable modification.

## Electron microscopy of RecA filamentation

A modified alcian method was used to visualize RecA filaments. Activated grids were prepared as described previously (*Lusetti et al., 2003b*). Samples for electron microscopy analysis were prepared as follows. All incubations were carried out at 37°C. Native RecA, RecAox, RecArep, RecAox, RecA^M164Q, or RecA^M35L (6.7 µM) was preincubated with 20 µM M13mp8 circular ssDNA or with ldsDNA in a 1× RecA buffer for 10 min. An ATP regeneration system (10 units/mL creatine phosphokinase and 12 mM phosphocreatine) was also included in the preincubation along with 1 mM DTT. ATP (3 mM) with or without 2 µM SSB protein was added, and the reaction was incubated for an additional 10 min after which ATPγS was added to 3 mM to stabilize the filaments and further incubated for an additional 3 min. In EM experiments involving linear dsDNA (lds), the experiment procedures were done without SSB proteins. In order to make sure that the filaments observed were not formed by RecA joining on its own, the control experiment was done without DNA, and no RecA filament was observed (data not shown).

The reaction mixture was diluted to a final DNA concentration of 0.4 ng/µL in 10 mM HEPES, 200 mM ammonium acetate, 10% glycerol (pH adjusted to 7.5), and adsorbed on alcian grid for 3 min. The grid was then touched to a drop of the above buffer followed by floating on a drop of the same buffer for 1 min. The sample was then stained by touching to a drop of 5% uranyl acetate followed by floating on a fresh drop of the same solution for 30 s. Finally, the grid was washed by touching to a drop of double-distilled water followed by immersion in two 10-mL beakers of double-distilled water. After the sample was dried, it was rotary-shadowed with platinum. This protocol is designed for visualization of complete reaction mixtures, and no attempt was made to remove unreacted material. Although this approach should yield results that give a true insight into reaction components, it does lead to samples with a high background of unreacted proteins. Imaging and photography were carried out with a Hitachi 7600 Transmission Electron Microscope equipped with a

**Table 1.** Sequences of RecA peptides containing Met residues and m/z expected for the different oxidation states.

| Peptide sequence | Reduced | 1× Ox | 2× Ox | 3× Ox |
| --- | --- | --- | --- | --- |
| MMSQAMR | 427.687 +2 | 435.684 +2 | 443.681 +2 | 451.678 +2 |
| AEIEGEIGDSHMGLAAR | 585.949 +3 | 591.280 +3 | – | – |
| IGVMFGNPETTTGGNALK | 903.956 +2 | 911.953 +2 | – | – |
| SMDVETISTGSLSLDIALGAGGLPMGR | 1324.667 +2 | 1332.664 +2 | 1340.661 +2 | – |

Hamamatsu ORCA HR camera. Digital images of the nucleoprotein filaments were taken at ×15,000 magnification except for SSB DNA molecules shown in the inset where the magnification is ×60,000. Imaging was also done using TECNAI G2 12 Twin Electron Microscope (FEI Co.) equipped with a 4k × 4k Gatan Ultrascan CCD camera at ×15,000 magnification.

For the measurement, filaments were randomly selected. For filament measurement on cssDNA, 10 circular filaments each from RecA wt + SSB, RecA wt No SSB, RecA M164Q + SSB, and RecA M164Q No SSB were selected. For filament measurement on ldsDNA, 54 and 53 linear filaments of respectively RecAwt and RecA M164Q were selected. In all experiments, filaments were measured three times using Metamorph analysis software. The average length of each filament was calculated in µm. The scale bar was used as a standard to calculate the number of pixels per µm.

## ATP hydrolysis assay

A coupled enzyme spectrophotometric assay was used to measure the DNA-dependent RecA ATPase activity as described (*Lindsley and Cox, 1990*). All assays were carried out using a Varian Cary 300 dual-beam spectrophotometer equipped with a temperature-controlled 12-cell changer. The cell path length is 1 cm, and the bandwidth was 2 nm. All reactions were carried out at 37°C in 1× RecA buffer, an ATP regeneration system (10 U/mL pyruvate kinase, 3 mM phosphoenolpyruvate), a coupled detection system (10 U/mL lactate dehydrogenase, 3 mM NADH), 3 mM ATP, 3 µM of RecA proteins, no or 0.5 µM of SSB protein, and 5 µM of M13mp18 cssDNA. Graphs represent the mean rate of the ATP hydrolysis, and errors represent the standard deviation from the biological triplicates. The apparent kcat was calculated as described previously (*Piechura et al., 2015*) by dividing the observed ATP hydrolysis rate by the number of RecA binding sites (i.e. the concentration in nucleotide divided by 3).

## DNA three-strand exchange

Three-strand exchange reactions were carried out in 1× RecA buffer as described (*Lusetti et al., 2003a*). All reactions contained an ATP regeneration system (10 unit/mL pyruvate kinase and 2.5 mM phosphoenolpyruvate). Reactions were incubated at 37°C. RecA proteins (3.5 µM) were preincubated with 10 µM M13mp18 circular ssDNA for 10 min. Then, SSB protein (1 µM) and 3 mM ATP were added and incubated for an additional 10 min. Reactions were initiated by the addition of M13mp18 linear dsDNA to 10 µM. Aliquot of 10 µL was removed at the indicated time points. The reaction was stopped by the addition of 5 µL of a solution 3:2 Ficoll:SDS. Samples were subjected to electrophoresis on 0.8% on Tris-acetate EDTA (TAE) agarose gel, stained with ethidium bromide, and visualized with Typhoon FLA-9000 (GE Healthcare). Experiments were run in triplicate, and results of a representative experiment are shown.

## Aerobiosis sensitivity assay

Cells were grown in LB in anaerobic condition until $OD_{600}$: 0.3–0.5. Then, they were serial diluted by a factor of 10 in Phosphate-buffered saline (PBS buffer), and 100 µL of the adequate dilution was spread in duplicate on LB plates and incubated at 37°C for 16 hr, one duplicate in anaerobic condition and the other in aerobic condition. Ampicillin was added to the culture and to the plates medium for complementation/suppression assay.

## UV sensitivity assay

Cells were grown in aerobic condition at 37°C until $OD_{600}$: 0.2–0.3 and serial diluted in PBS. Then, pure culture and dilutions from $10^{-1}$ to $10^{-6}$ were spotted on LB plates in several replicates. Replicated plates were exposed to increasing UV doses with a Crosslinker (Startalinker, Stargene). Plates were incubated 16 hr at 37°C, and survival cells were counted to determine the viability at different UV doses.

## Transduction assay

Transductant frequency assay with P1 phage was used as reporter of HR. P1 used for the assay was prepared as follows: P1 was grown on *lacZ::cat* MG1655 derivatives in soft agar plate and incubated 16 hr at 30°C. The top of the plate was collected, chloroform was added, and the mixture was vortexed and incubated at room temperature for 10 min. Agar and P1 were separated by

centrifugation, 10 min at 4500 rcf at 4°C. The upper layer containing the phage was collected and conserved at 4°C. P1 was serial diluted in PBS, and the titration of the phage was determined by spotting 10 μL of each dilution on MG1655 soft agar plates, the pfu/mL was between $10^{10}$ to $10^{11}$. For the assay, strains were grown in aerobic condition in LB until $OD_{600} \approx 1.0$. Also, 50 mL of cells were centrifuged 10 min at 4500 rcf. Pellets were washed with 50 mL of PMC buffer ($1\times$ PBS, 10 mM $MgSO_4$, and 5 mM $CaCl_2$) followed by a second centrifugation of 10 min. Pellets were resuspended in 5 mL of PMC. Then, 1 mL of cells ($\approx 2 \times 10^{10}$) was infected with the P1 phage ($\approx 10^8$, leading to a m.o.i. of around 0.005) and incubated at 37°C for 10 min without shaking. Also, 1 mL of non-infected cells was subject to the same protocol and used as negative control. Then, cells were centrifuged 3 min at 5500 rcf. Fresh LB supplemented with 10 mM Na citrate was used to resuspend cells pellets followed by 1 hr of incubation at 37°C with shaking. Then, 100 μL of cells was serial diluted in PBS and spotted on LB plates to determine the cfu/mL, and 100 μL of cells was spread on plates containing the adequate antibiotics. Experiments were at least replicated three times. The transductant frequency was calculated as follows: f – number of clones antibiotic-resistant per mL divided by the pfu per mL used for the infection. The differences and p-values of the transductant frequencies were determined by a Mann–Whitney statistical test. Values are represented in boxplot, and the median is indicated with a bold black line.

## MMC and oxygen growth capacity

LB agar plates containing or not 1 μg/mL MMC were prepared. MG1655 recA::spc strain harboring various versions of recA mutated plasmids was streaked onto each plate followed by overnight incubation at 37°C. The growth of each strain was checked by scoring for colony formation. Hpx⁻ recA::spc strain was transformed with the same plasmids as cited above and tested for its ability to grow in the presence or absence of oxygen.

## Microscopic examination

Microscopic examination of recA::spc strain cultures harboring various versions of recA mutated plasmids was realized using Motic BA310E 600 Biological Microscope. Light microscopy phase contrast imaging of E. coli cells in stationary phase cultured in LB were realized. E. coli cells carrying different recA mutated plasmids presented filamentous (RecA$^{M164Q}$) or normal (RecA, RecA$^{M35Q}$, RecA$^{M35L}$, RecA$^{M164L}$) morphologies. Experiments were run in triplicate, and results of a representative experiment are shown.

## SOS gene induction assay

Transcriptional fusion sfiA-lacZ was used as reporters of the SOS expression level. This construction was a gift from B. Michel (ΔlacZ::cat sfiA::Mud lacZ cat-amp) (Grompone et al., 2004). It is important to note that strains carrying this construction are mutated for sfiA as lacZ gene cassette disrupted the sfiA gene, as indicated in the Key Resources Table. Strains carrying the sfiA-lacZ reporter were grown in exponential phase $OD_{600}$:0.2–0.4 and exposed or not to 0.25 μg/mL MMC for 3 hr. The sfiA expression level was determined by β-galactosidase assays as described (Miller, 1972).

## Construction of recA-M164L allele in the chromosome

The chromosomal recA-M164L was generated by lambda red recombination adapted from Blank et al., 2011. The recA-M164L sce-I::kan DNA fragment was amplified by PCR using primers (5for recA M164L SceI K7kan/3rev recA K7Kan) with plasmid pWRG100 as a template. The resulting PCR product was electroporated into MG1655 cells after 1 hr of lambda recombinase expression from pWRG99 plasmid. After 1 hr of phenotypic expression, the cells were plated on LB containing 25 μg/mL kanamycin. Selected colonies contained a recA-M164L::sce-I::kan cassette, and harboring the pWRG99 plasmid has been newly isolated at 30°C. The kanamycin cassette was then removed by counter selection using lambda red recombination to insert a oligonucleotide recA hybrid of M164L-Scarless-f and M164L-Scarless-rv. The recA-M164L hybrid product was then electroporated into MG1655 recA-M164L::sce-I::kan expressing lambda recombinase from pWRG99, and after 1 hr of phenotypic expression serially diluted and plated on NA containing 100 μg/mL ampicillin and 1 μg/mL anhydrotetracycline (aTc). pWRG99 also encoded for the meganuclease I-Sce-I under the control

of an aTc-inducible promoter. The proper integration (positive Sce-I-resistant clones) of the recA-M164L mutational change was confirmed by diagnostic PCR and then sequenced.

## Culture competition with a two-color colony assay

Cell fitness was determined for each strain using a growth competition assay described by *Henrikus et al., 2019*. In general, this two-color colony assay is based on the color difference of Ara + and Ara− colonies on tetrazolium arabinose indicator plates (TA plates). Ara− colonies typically are red colored, while Ara+ colonies are white. Ara+ and Ara− cells can be counted, and thus fitness in a mixed population of two strains can be assessed. In preparation for the assay, individual over-night cultures of Ara− and Ara+ cells were grown in 3 mL of LB at 37°C in a 50-mL conical tube with an inclination of 90°. The next day a mixed culture of Ara− and Ara+ cells was set up at a 1:1 ratio by volume. To start the experiment, 3 mL of medium was inoculated with 30 μL of the mixed culture and grown at 37°C in a 50-mL conical tubes with an inclination of 90°. Fitness was assessed over the period of 120 hr; cells were serial diluted in PBS at 0, 24, 48, 72, 96, and 120 hr. The dilutions were spread on plates containing TA plates and incubated at 37°C for 16 hr before counting. We performed this assay-competing Ara+ cells of recA-M164L allele with Ara− cells of the recAWT and vice versa.

## Flow cytometry

To study the expression of *recA* in WT, Hpx⁻, and Hpx⁻ ΔmsrA ΔmsrB backgrounds, overnight bacterial cultures were diluted 1:100 in LB and incubated at 37°C for 5 hr until early stationary phase. For SOS response induction, overnight cultures were diluted 1:100 in LB and incubated at 37°C for 1 hr (early exponential phase). MMC (Sigma) was then added to a final concentration of 0.25 μg/mL, and the cultures were incubated for four additional hours (early stationary phase) at 37°C. Cells were then diluted in PBS. Data acquisition and analysis were performed using a Cytomics FC500-MPL cytometer (Beckman Coulter, Brea, CA). Data were collected for 100,000 events per sample and analyzed with CXP and FlowJo 8.7 software.

## *In vivo* LexA cleavage and RecA analysis by western blot

*E. coli* ΔrecA sfiA::lacZ strains transformed with plasmid carrying either WT RecA or RecA mutants were grown overnight. The next day subcultures were made and grown at 37°C until the $OD_{600}$ reached 0.2. Cultures were split, and 0.25 μg/mL MMC was added or not for each. The cultures were grown further at 37°C, and 1 mL of culture from each strain was aliquoted at 3 hr. The culture aliquots were washed once in cold PBS. The RecA protein levels were normalized to bacterial growth and resolved in 4–20% SDS-PAGE. The resolved bands were blotted to nitrocellulose membranes and probed with anti-LexA (1:5000) and anti-RecA (1:20,000) antibodies. Goat anti-rabbit IgG-Alk Phos (Chemicon International) and anti-rabbit HRP (Promega) were used as the secondary antibody at 1:10,000 dilutions to detect RecA and LexA proteins, respectively. Chemiluminescence detection was done using Amersham ECL western blotting kit and staining solution by using NBT/BCIP (Sigma). All the experiments were repeated at least three times for each RecA mutant, and representative result experiments are shown.

## Electrophoresis mobility shift assay

The ability of RecA proteins to bind DNA was assayed by EMSA. All reactions were realized in 1× RecA buffer supplemented with 1 mM DTT. Then, 20 μM of DNA, cssM13mp18 DNA, or pEAW951 nicked plasmid (cdsDNA) were incubated with RecA proteins at the indicated concentrations for 5 min at 37°C. Reactions were initiated by adding 3 mM of ATPγS followed by 30 min of incubation at 37°C. Reactions were stopped by adding 1:5 volume of glycerol EDTA dye (GED). Samples were submitted to electrophoresis in 0.8% TAE agarose gel, stained with ethidium bromide, and visualized with Typhoon FLA-9000 (GE Healthcare).

## Statistical analysis

Mann–Whitney U tests were performed using the QI-Macros software (KnowWare International, Inc, Denver, CO).

## Acknowledgements

We thank the members of the Barras, Ezraty, Cox, and Casadesus groups for comments on the manuscript, advice, and discussions throughout the work. CH was supported by the Fondation pour la Recherche Médicale (FRM-FDT20150532554) and by AMidex. This work was supported by grants from Agence Nationale Recherche (ANR) (# ANR-METOXIC), CNRS (#PICS-PROTOX), Aix-Marseille Université, Institut Pasteur, and the ANR-10-LABX-62-IBEID. Work from the Cox Lab was supported by grant GM32335 from the National Institute of General Medical Sciences (NIH). We thank the Wisconsin Veterinary Diagnostic Laboratory of the University of Wisconsin-Madison for allowing us to use their Hitachi 7600 TEM. We acknowledge the use of instrumentation supported by the UW MRSEC (DMR-1121288) and UW NSEC (DMR-0832760). We also thank M Modesti, B Michel, PL Moreau, C Jourlin-Castelli, L Espinosa, and JF Collet for helpful suggestions, reagents, and comments on the manuscript.

## Additional information

### Funding

| Funder | Grant reference number | Author |
|---|---|---|
| Agence Nationale de la Recherche | ANR-METOXIC | Benjamin Ezraty |
| Centre National de la Recherche Scientifique | PICS-PROTOX | Benjamin Ezraty |
| Agence Nationale de la Recherche | ANR-10-LABX-62-IBEID | Frédéric Barras |
| Fondation pour la Recherche Médicale | FRM - FDT20150532554 | Camille Henry |
| Aix-Marseille Université | AMidex | Camille Henry |
| National Institute of General Medical Sciences | GM32335 | Michael M Cox |

The funders had no role in study design, data collection and interpretation, or the decision to submit the work for publication.

### Author contributions

Camille Henry, Conceptualization, Data curation, Formal analysis, Supervision, Funding acquisition, Investigation, Writing - original draft, Project administration, Writing - review and editing; Laurent Loiseau, Resources, Data curation, Formal analysis, Investigation; Alexandra Vergnes, Resources, Data curation, Formal analysis; Didier Vertommen, Angela Mérida-Floriano, Sindhu Chitteni-Pattu, Elizabeth A Wood, Formal analysis; Josep Casadesús, Conceptualization, Data curation, Formal analysis, Funding acquisition, Writing - review and editing; Michael M Cox, Conceptualization, Data curation, Funding acquisition, Writing - review and editing; Frédéric Barras, Benjamin Ezraty, Conceptualization, Data curation, Formal analysis, Supervision, Funding acquisition, Writing - original draft, Project administration, Writing - review and editing

### Author ORCIDs

Camille Henry https://orcid.org/0000-0002-4011-6767
Alexandra Vergnes https://orcid.org/0000-0001-6745-6633
Angela Mérida-Floriano https://orcid.org/0000-0002-9650-7759
Josep Casadesús https://orcid.org/0000-0002-2308-293X
Michael M Cox http://orcid.org/0000-0003-3606-5722
Frédéric Barras https://orcid.org/0000-0003-3458-2574
Benjamin Ezraty https://orcid.org/0000-0003-3818-6907

### Decision letter and Author response

Decision letter https://doi.org/10.7554/eLife.63747.sa1

Author response https://doi.org/10.7554/eLife.63747.sa2

## Additional files

### Supplementary files
- Transparent reporting form

### Data availability

All data generated or analysed during this study are included in the manuscript and supporting files. Source data files have been provided in Dryad (https://doi.org/10.5061/dryad.zpc866t78).

The following dataset was generated:

| Author(s) | Year | Dataset title | Dataset URL | Database and Identifier |
|---|---|---|---|---|
| Henry C, Loiseau L, Vergnes A, Vertommen D, Mérida-Floriano A, Chitteni-Pattu S, Wood EA, Casadesús J, Cox MM, Barras F, Ezraty B | 2021 | Redox controls RecA protein activity via reversible oxidation of its methionine residues | http://dx.doi.org/10.5061/dryad.zpc866t78 | Dryad Digital Repository, 10.5061/dryad.zpc866t78 |

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

# Appendix 1

```
RecA-E.coli        ------------------------------------------------------------   0
RecA-bact          ------------------------------------------------------------   0
Rad51-H.sapiens    -------------------------------------------------MAMQMQLE      8
Rad51-S.cerevisiae MSQVQEQHISESQLQYGNGSLMSTVPADLSQSVVDGNGNGSSEDIEATNGSGDGGGLQEQ   60

RecA-E.coli        ------------------------------------------------------------   0
RecA-bact          ------------------------------------------------------------   0
Rad51-H.sapiens    ANADTSVEEES-----F-GPQPISRLEQCGINANDVKKLEEAGFHTVEAVAYAPKKELIN   62
Rad51-S.cerevisiae AEAQGEMEDEAYDEAALGSFVPIEKLQVNGITMADVKKLRESGLHTAEAVAYAPRKDLLE  120
                                                                   Met35
RecA-E.coli        ---AIDENKQKALAAALGQIEKQFGKGSIMRLGEDRSMDVETISTGSLSLDIALGAGGLP   57
RecA-bact          --VMSDEDKQKALEAALSQIEKQFGKGSIMRGLDKEAEDVEVISTGSLGLDIALGIGGLP   58
Rad51-H.sapiens    IKGISEAKADKILAEAAKLVPMGFTTAT---EFHQRRSEIIQITTGSKELDKLLQ-GGIE  118
Rad51-S.cerevisiae IKGISEAKADKLLNEAARLVPMGFVTAA---DFHMRRSELICLTTGSKNLDTLLG-GGVE  176
                              :  .:*  *    *    :    *  ..:      .  .  ::  ::***  **  *  **:
         Met58
RecA-E.coli        MGRIVEIYGPESSGKTTLTLQVIAA------AQREGKTCAFIDAEHALDPIYA----RKL  107
RecA-bact          RGRIIEIYGPESSGKTTLALHAIAE------AQKAGGVCAFIDAEHALDPVYA----KKL  108
Rad51-H.sapiens    TGSITEMFGEFRTGKTQICHTLAVTCQLPIDRGGGEGKAMYIDTEGTFRPERLLAVAERY  178
Rad51-S.cerevisiae TGSITELFGEFRTGKSQLCHTLAVTCQIPLDIGGGEGKCLYIDTEGTFRPVRLVSIAQRF  236
                     *  *  *::*    :**:  :          . :**:*  :: *          .:

RecA-E.coli        GVDIDNLLCS-------QPDTGEQALEICDALARSGAVDVIVVDSVAALTPKA-EIEGEI  159
RecA-bact          GVDIDNLLIS-------QPDTGEQALEIADMLVRSGAVDIIVVDSVAALVPKA-EIEGEM  160
Rad51-H.sapiens    GLSGSDVLDNVAYARAFNTDHQTQLLYQASAMMVESRYALLIVDSATALYRTDYSGRGEL  238
Rad51-S.cerevisiae GLDPDDALNNVAYARAYNADHQLRLLDAAAQMMSESRFSLIVVDSVMALYRTDFSGRGEL  296
                     *:.  .:  *  .          :  *    :  *    .  :   ..   :::***. **   .  .  .**:
                           Met164                                    Met202
RecA-E.coli        GDSHMGLAARMMSQAMRKLAGNLKQSNTLLIFINQIRMK--IGVMF-GNPETTTGGNALK  216
RecA-bact          GDSHVGLQARLMSQALRKLTGSISKSNTTVIFINQIRMK--IGVMF-GNPETTTGGNALK  217
Rad51-H.sapiens    SARQMHLAR-----FLRMLLRLADEFGVAVVITNQVVAQVDGAAMFAADPKKPIGGNIIA  293
Rad51-S.cerevisiae SARQMHLAK-----FMRALQRLADQFGVAVVVTNQVVAQVDGGMAFNPDPKKPIGGNIMA  351
                     .  ::  *          :*  *     .:  .. ::.  **:    :      *    :*:.   *** :

RecA-E.coli        FYASVRLDIRRIGAVKEGENVVGSETRVKVVKNKIAAPFKQAEFQILYGEGINFYGELVD  276
RecA-bact          FYASVRLDIRRIGSIKDGDEVIGNRTRVKVVKNKVAPPFKQAEFDIMYGEGISREGELID  277
Rad51-H.sapiens    HASTTRLYLRKGRGE----------TRICKIYDSPCLPEAEAMFAI-NADGVGDAKD---  339
Rad51-S.cerevisiae HSSTTRLGFKKGKGC----------QRLCKVVDSPCLPEAECVFAI-YEDGVGDPREEDE  400
                     .  ::.** ::: .               *:  :  :.  .  *  :.  *  *    :*:.    :

RecA-E.coli        LGVKEKLIEKAGAWYSYKGEKIGQGKANATAWLKDNPETAKEIEKKVRELLLSNPNSTPD  336
RecA-bact          LGVKLGIVEKSGAWYSYNGEKICQGRENAKQYLKENPELAEEIEKKIREKLGLSSSAAAS  337
Rad51-H.sapiens    ------------------------------------------------------------  339
Rad51-S.cerevisiae ------------------------------------------------------------  400

RecA-E.coli        FSVDDSEGVAETNEDF  352
RecA-bact          ETDEDSEEEEEAEE--  351
Rad51-H.sapiens    ----------------  339
Rad51-S.cerevisiae ----------------  400
```

**Appendix 1—figure 1.** Amino acid sequence alignment of RecA/Rad51. The protein sequence comparison between the bacterial RecA consensus sequence (RecA-*bact*) obtained from *Lusetti and Cox, 2002*, the *E. coli* RecA sequence (RecA-*E. coli*), and the human and *S. cerevisiae* Rad51 sequences (Rad51-*H. sapiens* and Rad51-*S. cerevisiae*) was performed using Clustal Omega (https://www.ebi.ac.uk/Tools/msa/clustalo/). Methionine residues (Met35, Met58, Met164, Met202) are colored in red above the sequence. * indicates positions that have a single, fully conserved residue, : indicates conservation between groups of strongly similar properties, and . indicates conservation between groups of weakly similar properties.

