## [Decision Letter]

[Editors’ note: the authors submitted for reconsideration following the decision after peer review. What follows is the decision letter after the first round of review.]

Thank you for submitting your work entitled "Homeostasis of the DNA repair RecA protein is under redox control" for consideration by *eLife*. Your article has been reviewed by three peer reviewers, and the evaluation has been overseen by a Reviewing Editor and a Senior Editor. The reviewers have opted to remain anonymous.

Our decision has been reached after consultation between the reviewers. Based on these discussions and the individual reviews below, we regret to inform you that your work will not be considered further for publication in *eLife*.

This is an interesting paper but it has some holes, especially regarding the *in vivo* significance. It is likely that msrAB mutants are pleiotropic, affecting other recombination and repair proteins. The findings are suggestive, but not sufficiently compelling. Collectively, the reviewers and editor did not think that another 2-months of work by the authors would result in a paper that was likely to be publishable. Much more work is needed, and all felt that though interesting, the conclusions were premature based on the evidence presented.

One experiment that I was looking for was use of a RecA that had all methionine residues substituted to non-oxidaizeble variants: this mutant, in principle, would be largely insensitive to mutation of msrAB.

Reviewer #1:

This manuscript describes experiments showing that the persistence of oxidized methionine residues in RecA protein, in an Hpx msrA msrB mutant, interfere with its ability to carry out DNA repair, π transduction or induce the SOS regulon. This is supported by in vitro experiments showing that oxidized RecA protein is defective for DNA strand exchange, DNA-dependent ATP-hydrolysis and formation of extended filaments on ssDNA.

Inactivation of all the tested activities of oxidized RecA *in vivo* and *in vitro* are recapitulated by RecAM35Q mimicking oxidation of RecA M35. This is not unexpected given the results described above. Interestingly by contrast, RecAM165Q is defective for DNA repair but is constitutively induced for SOS. Consistent with this, RecAM165Q protein can form filaments on ssDNA and stimulate DNA-dependent ATPase even in the absence of SSB suggesting that it has enhanced filament propagation capacity. RecAM165Q protein is however unable to promote strand exchange, explaining the defect in DNA repair.

The authors argue that this may provide a way of regulating the cell's response to oxidative DNA damage. If RecA M165 is oxidized this will result in the induction of the SOS response. MsrA and MsrB can then reverse this oxidation and allow DNA repair to occur. However, the problem is that there is no evidence that this happens. There are 4 methionine residues in RecA and there is no evidence that M165 is preferentially oxidized. If the other residues are oxidized, then RecA may simply not function at all (as for M35). Furthermore, if RecA M164 has evolved to respond to oxidation by activating the SOS response, there is no reason to argue why it has not evolved to repair DNA in the oxidized state. It seems to me that the authors are seeing varying degrees of RecA inactivation by oxidation of different residues that may not imply anything more than partial or complete inactivation without any functional meaning.

Reviewer #2:

The paper describes experiments that attempt to correlate the oxidation of specific Met residues in the RecA protein with its *in vivo* functions in SOS induction, DNA repair and recombination and its activities to bind ssDNA, ATPase activity and in vitro recombination. They argue that increased sensitivity to O2 and DNA damaging agents in MsrAB- Hpx- mutant occurs because RecA is oxidized at certain Met residues and unable to perform its function. They first focus on the in vitro studies where they show that the Met residues are oxidized and that the oxidized RecA fails to display a number of RecA activities in vitro. This oxidized form of RecA can be partially rescued by MsrAB treatment. They then show that HpX MsrAB mutants are unable to do P1 mediated recombination, UV repair, SOS induction and cell growth as well as wild type (but not nearly as defective as a RecA deleted strain where tested). Then they zero in on two residues M35 and M164 and making four mutants with both of these codons changed to Gln(Q) that is supposed to be a mimic of the oxidation state of Met and Leu(L) as a mutant control. They then characterized some of them in several assays. They find that any change to M35 results in a defective protein *in vivo* and in vitro and that changes at M164 provided more interesting behavior. M164Q turns out to be SOS constitutive *in vivo* and can bind ssDNA better than wt in the presence of SSB. This behavior is reminiscent of RecA730 but this protein is Rec- *in vivo* and in vitro for all assays tested. The M165L protein is resistant to MMC and able to induce SOS but is Rec- *in vitro* and has no ssDNA stimulated ATPase activity in vitro (supplementary results). The idea that they would like to have the reader come away with is that the oxidation damage in the cell will modify the residue on RecA at position M164 and this will in turn help induce the SOS response and then this damage will be reversed by by the action of MsrAB so that RecA can do its job in recombinational DNA repair.

This is a very exciting paper and yields a very satisfying mechanism to explain the results. That being said, I have some concerns about the paper before I think it will be ready for publication.

1) in vivo, the treatments oxidize many proteins in the cell, not just RecA. In fact, do we know that RecA is oxidized in vivo at the Met residues? It would be nice to show.

2) The phenotypes that the authors show deficiency in recombination, cell growth etc under the oxidized conditions could be due to the oxidation of many different proteins. The one that comes to mind first is RecBCD (It would be Rec-, UV sens, poor growth, poor SOS induction with some treatments). This is particularly a concern because the decreases show in UV sensitivity and P1 mediated recombination are small and could be due to the oxidation of other protein(s), not RecA. Do they have an experiment that shows it is the oxidation of RecA alone that is causing the phenotypes.

3) Another concern with the in vivo results is the "complementation" data with the RecA mutants on plasmids. How do we know that their phenotypic effect, the complementation (or lack thereof with the mutants), is not due to overproduction of a mutant protein.

4) The Materials and methods do not describe their expression system well enough. Figure 3 does have a western blot, and we see that there is much more recA expressed from the plasmid then the empty vector. I would like to see their most interesting mutant, RecA M164Q, put on the chromosome and characterized for their phenotypes. In my opinion, this would give the work much more significance and impact. Expression from multi-copy plasmids is open to multiple interpretations. I am confused by the observation that the RecA M164L mutant is MMC resistance and can induce SOS but yet cannot do recombination in vitro (supplementary results) or in vivo? (has this been tested?). Do the authors have an explanation for this?

5) In the MsrAB rescue in vitro experiments, why does the RecA ATPase not come back to full activity? If the authors say that only the Met residues are oxidized and the MsrAB can reverse this damage, then why is full activity not restored? Is some other residue(s) being oxidized?

6) It would seem that RecA as isolated from the cell has about 20% of its Met residues (35,59,164 and 202) oxidized. Is this happening in vivo or as part of the isolation process. I think it needs comment as it affects the interpretation of the results.

Reviewer #3:

The biology of Hpx- and msr mutants is potentially interesting, but needs more work. A lot more, I suspect. The biochemistry of mutant RecA proteins, even though done professionally, is too predictable. As a result, the authors cannot claim that part of the acute oxidative assault on the cell is a selective inactivation of key DNA repair enzymes. However, it could be one of the (minor) consequences of chronic hydrogen peroxide exposure – this remains to be explored.

Treatment with 50 mM hydrogen peroxide for 2 hours at 37{degree sign}C is not biologically relevant. In fact, 50 mM hydrogen peroxide kills most type of cells, including *E. coli*, on contact. More biologically relevant would be isolation of RecA from Hpx- msr mutant cells to check its in vivo modifications. However, I would be worried about mass-spec detection, due to the known high methionine oxidation background of the procedure.

It is surprising that paper talking about oxidative damage has not employed a single acute oxidative treatment (like superoxide, or hydrogen peroxide or, perhaps the most relevant for protein oxidation, hypochlorous acid).

The paper does utilize chronic hydrogen peroxide stress, but Hpx- mutants should not be too sensitive to acute treatments with μM concentrations of hydrogen peroxide. Moreover, msr mutants in Hpx+ background could be treated with millimolar concentrations. Have the authors tried those for hydrogen peroxide sensitivity at all?

The authors agree that all their in vivo phenotypes could be economically explained by a defect in SOS-induction. The authors propose the SOS defect is due to dysfunctional RecA filament. However, equally possible, is the SOS defect due to uncleavable LexA repressor.

[Editors’ note: further revisions were suggested prior to acceptance, as described below.]

Thank you for submitting your article "Redox controls RecA protein activity via reversible oxidation of its methionine residues" for consideration by *eLife*. Your article has been reviewed by two peer reviewers, and the evaluation has been overseen by a Reviewing Editor and Gisela Storz as the Senior Editor. The reviewers have opted to remain anonymous.

The reviewers have discussed the reviews with one another and the Reviewing Editor has drafted this decision to help you prepare a revised submission.

Summary:

This is an important paper that describes the role of protein damage, specifically damage of RecA protein, by DNA damaging agents, ROS, in a DNA damage response by bacteria. It report specific mechanistic details about how oxidation alters the function of RecA protein and the physiological consequences. By using a traditional DNA imaging agent but focusing on protein damage, this work establishes an important consequence of protein damage (oxidation of methionines) and the importance of protein-damage repair (sulfhydryl reduction) processes.

Revisions:

This manuscript should be accepted, but needs an important major revision. Reviewer #2 provided some valid critiques of the work: "The authors chose to use in-gel digestion followed by mass spectrometry to determine which Met were oxidized in cells exposed to HOCl….Yes, it is common, but what has not been recognized is that the level is totally artifactual. In gel digestion has been clearly documented to lead to cause absurdly high levels of MetO. See, for example, "Assessment of Sample Preparation Bias in Mass Spectrometry-Based Proteomics", Anal Chem 90: 5405 2018, especially Figure 3D."

Consequently, the authors need include the following reservation or equivalent:

The authors need to point out in the manuscript the problems with their approach to the mass spectrometry, most notably the methionine oxidation artifact associated with their in-gel digestion (cite the Anal. Chem. reference). After noting the difficulties with artifactual Met oxidation with mass spec, they might simply say that they used the in vitro results to guide their in vivo studies."

---

## [Author Response]

[Editors’ note: the authors resubmitted a revised version of the paper for consideration. What follows is the authors’ response to the first round of review.]

This is an interesting paper but it has some holes, especially regarding the in vivo significance. It is likely that msrAB mutants are pleiotropic, affecting other recombination and repair proteins. The findings are suggestive, but not sufficiently compelling. Collectively, the reviewers and editor did not think that another 2-months of work by the authors would result in a paper that was likely to be publishable. Much more work is needed, and all felt that though interesting, the conclusions were premature based on the evidence presented.

Indeed more than 2-months was needed to collect new data that strengthen our conclusion. We have now collected new evidences, which we think strengthen the work and further support our first findings and interpretations. One of the issue was to know whether RecA is oxidized in vivo. We did obtain evidences for this by using gel-shift assay under HOCl stress (see Figure 3B). This is a key finding as it validates and backs up nicely all of the other interpretations of both in vivo and in vitro observations.

One experiment that I was looking for was use of a RecA that had all methionine residues substituted to non-oxidaizeble variants: this mutant, in principle, would be largely insensitive to mutation of msrAB.

This prediction is valid but there is no way to test it, except in vitro, as the sole substitution of Met35 to the non-oxidizable Leu residue is enough to inactivate RecA. Mutating all of the Met residues would not add much and it would be impossible to evaluate the functional effect of the lack of oxidation or absence of repair system.

Reviewer #1:This manuscript describes experiments showing that the persistence of oxidized methionine residues in RecA protein, in an Hpx msrA msrB mutant, interfere with its ability to carry out DNA repair, π transduction or induce the SOS regulon. This is supported by in vitro experiments showing that oxidized RecA protein is defective for DNA strand exchange, DNA-dependent ATP-hydrolysis and formation of extended filaments on ssDNA.Inactivation of all the tested activities of oxidized RecA in vivo and in vitro are recapitulated by RecAM35Q mimicking oxidation of RecA M35. This is not unexpected given the results described above. Interestingly by contrast, RecAM165Q is defective for DNA repair but is constitutively induced for SOS. Consistent with this, RecAM165Q protein can form filaments on ssDNA and stimulate DNA-dependent ATPase even in the absence of SSB suggesting that it has enhanced filament propagation capacity. RecAM165Q protein is however unable to promote strand exchange, explaining the defect in DNA repair.The authors argue that this may provide a way of regulating the cell's response to oxidative DNA damage. If RecA M165 is oxidized this will result in the induction of the SOS response. MsrA and MsrB can then reverse this oxidation and allow DNA repair to occur. However, the problem is that there is no evidence that this happens. There are 4 methionine residues in RecA and there is no evidence that M165 is preferentially oxidized. If the other residues are oxidized, then RecA may simply not function at all (as for M35). Furthermore, if RecA M164 has evolved to respond to oxidation by activating the SOS response, there is no reason to argue why it has not evolved to repair DNA in the oxidized state. It seems to me that the authors are seeing varying degrees of RecA inactivation by oxidation of different residues that may not imply anything more than partial or complete inactivation without any functional meaning.

We shared both reviewer’s concerns: (i) the biochemical basis of differential level of oxidation of the Met residues and (ii) the teleonomic issue about the putative evolutionary advantage of a specific role of M164. Two new experiments were carried out that relate to these questions and modified our view and conclusion.

First, we performed a mass-spec analysis of RecA subjected to increasing concentration of HOCl in order to have a dose/response sensitivity of Met residues to oxidation. We observed that M164 is more prone to oxidation than Met35. This result is now included in the new modified version (Figure 7D). Clearly this “enhanced” sensitivity to M164 is consistent with the hypothesis we developed in the previous version, according to which some molecules of RecA could be oxidized solely on Met164 and fires on the SOS response to permit a anticipatory strategy. Second, we did experiments aimed at assessing the physiological advantage that the presence of an oxidizable Met at position 164 could provide. A strain producing RecA-M164Q which mimics a RecA molecule locked in a oxidized state at position 164 showed a very slow growth and filamentous phenotype. Moreover, we run a competition assay between the wild type and the strain synthesizing the RecA-M164L protein. Surprisingly, the RecA-M164L synthesizing strain showed enhanced fitness and this enhanced fitness was dependent upon aerobic conditions.

Thus these new results do show that Met residues exhibit diverse sensitivity towards oxidation and M164 is the most sensitive, as we speculated it could be in our first version. However, growth experiments showed that, M164 oxidation is deleterious for bacterial fitness. All of the data of the fitness experiments are presented in the text and Figure 6. Therefore, we do no longer present the speculative view of an anticipatory effect of oxidation at position 164 and rather conclude that ROS can inactivate RecA, the level of which can be replenished by *E. coli* in multiple ways as depicted in Figure 8.

Reviewer #2:The paper describes experiments that attempt to correlate the oxidation of specific Met residues in the RecA protein with its in vivo functions in SOS induction, DNA repair and recombination and its activities to bind ssDNA, ATPase activity and in vitro recombination. They argue that increased sensitivity to O2 and DNA damaging agents in MsrAB- Hpx- mutant occurs because RecA is oxidized at certain Met residues and unable to perform its function. They first focus on the in vitro studies where they show that the Met residues are oxidized and that the oxidized RecA fails to display a number of RecA activities in vitro. This oxidized form of RecA can be partially rescued by MsrAB treatment. They then show that HpX MsrAB mutants are unable to do P1 mediated recombination, UV repair, SOS induction and cell growth as well as wild type (but not nearly as defective as a RecA deleted strain where tested). Then they zero in on two residues M35 and M164 and making four mutants with both of these codons changed to Gln(Q) that is supposed to be a mimic of the oxidation state of Met and Leu(L) as a mutant control. They then characterized some of them in several assays. They find that any change to M35 results in a defective protein in vivo and in vitro and that changes at M164 provided more interesting behavior. M164Q turns out to be SOS constitutive in vivo and can bind ssDNA better than wt in the presence of SSB. This behavior is reminiscent of RecA730 but this protein is Rec- in vivo and in vitro for all assays tested. The M165L protein is resistant to MMC and able to induce SOS but is Rec- in vitro and has no ssDNA stimulated ATPase activity in vitro (supplementary results). The idea that they would like to have the reader come away with is that the oxidation damage in the cell will modify the residue on RecA at position M164 and this will in turn help induce the SOS response and then this damage will be reversed by by the action of MsrAB so that RecA can do its job in recombinational DNA repair.This is a very exciting paper and yields a very satisfying mechanism to explain the results. That being said, I have some concerns about the paper before I think it will be ready for publication.1) in vivo, the treatments oxidize many proteins in the cell, not just RecA. In fact, do we know that RecA is oxidized in vivo at the Met residues? It would be nice to show.

We understand and share the reviewer’s frustration. However, the only way to get this piece of information (i.e. that Met residues are oxidized in vivo) would be by using mass spectrometry analysis and this proved to be non-reliable (see answer to reviewer 3 below). The only answer we can give is that we obtained additional confidence in the oxidation of RecA in vivo by applying gel shift approach. This approach is based on the fact that proteins oxidized on their Met residues show a slower migration profile than the reduced one. This was used in several previous studies in the field including in our previous work on MsrP (Gennaris et al., 2015). These data are crucial to our study and they have now been included in the modified new version (Figure 3B). Also we were able to show that indeed in vivo after an HOCl stress, RecA is more oxidized in a *msrAB* mutant as compared to the wt. The fact that the absence of MsrAB does show a consequence on RecA migration does support the notion that Met residues were oxidized in the first place. And we might remind that in vitro even we used non-physiological large amounts of H2O2, we had no evidences for oxidation of residues other than Met. Taken together, it seems to us that these series of converging data do indicate that RecA Met residues are oxidized upon exposure to oxidative stress.

2) The phenotypes that the authors show deficiency in recombination, cell growth etc under the oxidized conditions could be due to the oxidation of many different proteins. The one that comes to mind first is RecBCD (It would be Rec-, UV sens, poor growth, poor SOS induction with some treatments). This is particularly a concern because the decreases show in UV sensitivity and P1 mediated recombination are small and could be due to the oxidation of other protein(s), not RecA. Do they have an experiment that shows it is the oxidation of RecA alone that is causing the phenotypes.

We agree with the reviewer and accordingly investigated whether other recombination/repair associated proteins could actually be the targets of ROS. We attempted to test whether overproduction of some of them such as RecR or UvrB, could act as multicopy suppressors. This would have been an indication ROS would have reduce their functional concentration. Unfortunately, overexpression of RecR and UvrB turned out to be toxic for Hpx *E. coli* and we could not get reliable data. Thus, although we cannot rule out the possibility that multiple proteins might be damaged and as a whole contribute to the observed phenotype, we feel we provide sufficient evidence, both genetic and biochemical, to support the idea that at least RecA is one of them and that its oxidation contributes significantly to the UV sensitivity of the Hpx *msrAB uvr*A strain. However, we thank the reviewer for pointing out the possibility of other hits and we introduced a sentence explicating this fact.

3) Another concern with the in vivo results is the "complementation" data with the RecA mutants on plasmids. How do we know that their phenotypic effect, the complementation (or lack thereof with the mutants), is not due to overproduction of a mutant protein.

We agree with reviewer. See below for our answer.

4) The Materials and methods do not describe their expression system well enough. Figure 3 does have a western blot, and we see that there is much more recA expressed from the plasmid then the empty vector. I would like to see their most interesting mutant, RecA M164Q, put on the chromosome and characterized for their phenotypes. In my opinion, this would give the work much more significance and impact.

We followed the reviewer’s suggestion, and we put the most interesting alleles (recAM164Q and recAM164L) on the chromosome. Satisfyingly all the phenotypes observed with these new strains recapitulated those observed previously by using the plasmid expression system (Figure 5).

Expression from multi-copy plasmids is open to multiple interpretations. I am confused by the observation that the RecA M164L mutant is MMC resistance and can induce SOS but yet cannot do recombination in vitro (supplementary results) or in vivo? (has this been tested?). Do the authors have an explanation for this?

It appears that the reviewer might have made confusion between RecA-M164L and RecAM35L. Indeed, in vivo the M164L encoding strain exhibits wild-type like MMC resistance and SOS induction levels but we didn’t perform any biochemical characterization of this variant. The supplementary results (Figure 5) shows biochemical characterization of the M35L variant, which is affected in all activities and in vivo lead to an inactive RecA (MMC sensitive and non-inducible SOS) (Figure 4). We hope the changes introduced (Figure 4 C, D, E, F and Figure 5) clarified this point.

5) In the MsrAB rescue in vitro experiments, why does the RecA ATPase not come back to full activity? If the authors say that only the Met residues are oxidized and the MsrAB can reverse this damage, then why is full activity not restored? Is some other residue(s) being oxidized?

The reviewer is right. None of the RecA activities are full restored (ATPase, nucleoprotein filaments formation and strand exchange) following MsrAB repair. Mass-spec analysis revealed no other modification (i.e. Cys modification, R-K-P-T carbonylation ). One possibility is that full recovery requires a chaperon. Such a hypothesis was put forward in studies on catalase oxidation/repair in *Helicobacter pylori* (Mahawar, 2011, doi: 10.1074/jbc.M111.223677).

6) It would seem that RecA as isolated from the cell has about 20% of its Met residues (35,59,164 and 202) oxidized. Is this happening in vivo or as part of the isolation process. I think it needs comment as it affects the interpretation of the results.

The presence of a basal level (around 20%) of oxidized Met in protein is common in the field. We setup an anaerobic purification protocol of protein and even with an anaerobic growth culture we got Met-O presence in agreement with the reviewer #3 comments about the “high methionine oxidation background of the procedure”.

Reviewer #3:The biology of Hpx- and msr mutants is potentially interesting, but needs more work. A lot more, I suspect. The biochemistry of mutant RecA proteins, even though done professionally, is too predictable. As a result, the authors cannot claim that part of the acute oxidative assault on the cell is a selective inactivation of key DNA repair enzymes. However, it could be one of the (minor) consequences of chronic hydrogen peroxide exposure – this remains to be explored.Treatment with 50 mM hydrogen peroxide for 2 hours at 37{degree sign}C is not biologically relevant. In fact, 50 mM hydrogen peroxide kills most type of cells, including *E. coli*, on contact. More biologically relevant would be isolation of RecA from Hpx- msr mutant cells to check its in vivo modifications. However, I would be worried about mass-spec detection, due to the known high methionine oxidation background of the procedure.

We agree that 50 mM H2O2 is not biologically relevant; this treatment was only used for testing whether in vitro RecA could be oxidized and whether Met residues would be targets of H2O2. We have now completed the oxidation experiments by submitting RecA to µM concentration of HOCl as Met residues are known to be highly sensitivity to HOCl.

We set up different protocols of purification of RecA (aerobic or anaerobic) and produce it in different strains (wt, msrAB, Hpx, Hpx msrAB) which were exposed or not to HOCl. Eventually mass spec analysis was carried out to characterize oxidation state of the resulting purified and treated RecA samples mass-spec analysis. Unfortunately, and as anticipated by this reviewers 3, results from several different experiments lack reproducibility. In particular we noticed that the level of spontaneous oxidation, even in sample prepared anaerobically, was very high (approx. 20%) and also we noticed that depending upon the background used, RecA formed high aggregates. Although these aggregates could be sign of misfolding due to various level of oxidation, this essentially introduced great variability in the purification protocol, which eventually could have misled us. Therefore, we are sorry and very frustrated about this given the amount of efforts we put in this approach, but we cannot but give up use of mass spec analysis to characterize the oxidation status of RecA in vivo.

It is surprising that paper talking about oxidative damage has not employed a single acute oxidative treatment (like superoxide, or hydrogen peroxide or, perhaps the most relevant for protein oxidation, hypochlorous acid).

We thank the reviewer for this excellent idea and we followed his suggestion. The *msrAB* mutant was treated with HOCl and by using a gel shift assay, we were able to show that absence of MsrAB causes an enhanced level of oxidized RecA in vivo (Figure 3B).

The paper does utilize chronic hydrogen peroxide stress, but Hpx- mutants should not be too sensitive to acute treatments with μM concentrations of hydrogen peroxide. Moreover, msr mutants in Hpx+ background could be treated with millimolar concentrations. Have the authors tried those for hydrogen peroxide sensitivity at all?

Although one can find in the literature reports about sensitivity of *msr* mutant to H2O2 (St John et al., 2001 ; Zhao et al., 2010 ; Denkel et al., 2011), our laboratory has never been able to observe such a phenotype in *E. coli*.

The authors agree that all their in vivo phenotypes could be economically explained by a defect in SOS-induction. The authors propose the SOS defect is due to dysfunctional RecA filament. However, equally possible, is the SOS defect due to uncleavable LexA repressor.

The reviewer is right and such a possibility could indeed be envisioned. It would require three predictions. (i) LexA is oxidized on its Met residues, which is entirely possible, (ii) oxidation of LexA made it more resistant to RecA mediated cleavage and (iii) overexpression of RecA would overcome the intrinsic resistance of oxidized LexA. All of these situations are indeed possible, yet they seem to require much more speculative assumptions than the more economic scenario we propose here. In fact LexA was indeed pointed out as a target of ROS in the literature. However, the outcome was opposite to what is predicted in the model above as oxidized LexA was no longer able to repress SOS system, resulting in full expression of SOS (Schook et al., 2011). This is cited in our version. Furthermore we have never been able to observe a difference on LexA motility by using gel-shift assay with antiLexA antibodies. Yet, in agreement with reviewer 2 comment #2, we made explicit the fact that we cannot exclude that other DNA repair/recombination proteins were also hit by oxidative stress.

[Editors’ note: what follows is the authors’ response to the second round of review.]

Revisions:This manuscript should be accepted, but needs an important major revision. Reviewer #2 provided some valid critiques of the work: "The authors chose to use in-gel digestion followed by mass spectrometry to determine which Met were oxidized in cells exposed to HOCl….Yes, it is common, but what has not been recognized is that the level is totally artifactual. In gel digestion has been clearly documented to lead to cause absurdly high levels of MetO. See, for example, "Assessment of Sample Preparation Bias in Mass Spectrometry-Based Proteomics", Anal Chem 90: 5405 2018, especially Figure 3D."Consequently, the authors need include the following reservation or equivalent:The authors need to point out in the manuscript the problems with their approach to the mass spectrometry, most notably the methionine oxidation artifact associated with their in-gel digestion (cite the Anal. Chem. reference). After noting the difficulties with artifactual Met oxidation with mass spec, they might simply say that they used the in vitro results to guide their in vivo studies."

Indeed, results shown in Figure 3A provide from in-gel digestion based analysis to characterize Met residues targeted by H2O2 and reduced by MsrA/B. We thank reviewer #2 to underline the risk of Met oxidation during ingel digestion as reported in the study of Klont et al., 2018. We wish to make clear however that all the samples (native RecA, RecAox and RecArep) have been run and digested at the same time and in the same way. So the difference in MetO content observed between the different forms is likely to reflect intrinsic features of each of them. Taking into account the reviewer’s comment, we conclude that if anything else, this experiment might have led to an over estimation of MetO in all samples and has little chance to have introduced artefactual biases in the comparison between samples.

Moreover, the data presented in the new Figure 8D (formerly Figure 7D) were obtained by treating the sample with HOCl, and not H2O2, and without using gel separation. Results obtained in the absence of treatment (no H2O2, no HOCl) gave the same order of magnitude of basal MetO content, i.e. around 20%. Therefore, we feel confident that in our experimental setup (Figure 3A) the overvalued MetO content due to in-gel digestion seems to be very low.

Nonetheless, the reviewer is right and the risk associated with in gel digestion is a potential source of artifact that needs to be pointed out. Therefore, we underlined it in the new version of the manuscript wherein the possibility of artefact Met oxidation during in-gel digestion is made explicit along with the reference to the Klont et al., 2018 paper. As suggested, we warn the reader of these potential pitfalls, yet we explain how these in vitro results were helpful in guiding us for in vivo studies and helping us to choose which Met residues should be further studied, i.e.

M35 and M164.

These modifications have now been included in the modified new version.

Moreover, as suggested by reviewer #2, we refer to the Metosite database, which provides data of sulfoxidation sites in proteins.